# Malaria parasite centrins can assemble by Ca$^{2+}$-inducible condensation

**Yannik Voß[1,2], Severina Klaus[1,3], Nicolas P. Lichti[1], Markus Ganter[1], Julien Guizetti[1] ***

**1** Center for Infectious Diseases, Heidelberg University Hospital, Heidelberg, Germany, **2** German Center for Infection Research, partner site Heidelberg, Heidelberg, Germany, **3** Department of Infectious Diseases, Virology, Heidelberg University Hospital, Heidelberg, Germany

\* julien.guizetti@med.uni-heidelberg.de

**Data Availability Statement:** All data is provided in the manuscript or supplemental information. Image analysis macros are deposited on https://github.com/SeverinaKlaus/ImageJ-Macros.

## Abstract

Centrins are small calcium-binding proteins that have a variety of roles and are universally associated with eukaryotic centrosomes. Rapid proliferation of the malaria-causing parasite *Plasmodium falciparum* in the human blood depends on a particularly divergent and acentriolar centrosome, which incorporates several essential centrins. Their precise mode of action, however, remains unclear. In this study calcium-inducible liquid-liquid phase separation is revealed as an evolutionarily conserved principle of assembly for multiple centrins from *P. falciparum* and other species. Furthermore, the disordered N-terminus and calcium-binding motifs are defined as essential features for reversible biomolecular condensation, and we demonstrate that certain centrins can form co-condensates. In vivo analysis using live cell STED microscopy shows liquid-like dynamics of centrosomal centrin. Additionally, implementation of an inducible protein overexpression system reveals concentration-dependent formation of extra-centrosomal centrin assemblies with condensate-like properties. The timing of foci formation and dissolution suggests that centrin assembly is regulated. This study thereby provides a new model for centrin accumulation at eukaryotic centrosomes.

## Author summary

Malaria occurs when a small parasite, named *Plasmodium falciparum*, invades our red blood cells where it grows and multiplies. Parasite multiplication requires segregation of the replicated genetic material into multiple new nuclei, which are then packaged into multiple new daughter cells at once. This process is coordinated by a small cellular structure called centrosome. The centrosome of malaria parasites remains poorly characterized, but centrin proteins are one of the few conserved components. In this study we investigate the properties of centrins that were thought to assemble by polymerization. We, however, discovered that centrins can assemble by so-called liquid-liquid phase separation. Phase separation can be conceptualized like oil droplets floating on a soup, but in a biological context the droplets are highly concentrated in a specific protein while the surroundings are largely depleted. We uncover that specific structural features and calcium binding are required for centrins to rapidly undergo phase separation. To investigate

**Funding:** This work was supported by the German Research Foundation (DFG) 349355339 to J.G, the Human Frontiers Science Program (HFSP) CDA00013/2018-C to J.G, the Chica and Heinz Schaller Foundation to J.G, the Studienstiftung des Deutschen Volkes to Y.V, The German Research Foundation (DFG) 240245660 - SFB 1129 to M.G. The funders had no role in study design, data collection and analysis, decision to publish, or preparation of the manuscript.

**Competing interests:** The authors have declared that no competing interests exist.

centrin behavior in vivo we developed a novel expression system to modulate protein concentration, which is a critical feature of phase separation. Our study provides a new model of how centrins might accumulate at the centrosome and reveals a previously unknown property of this highly conserved protein family.

## Introduction

Malaria-causing parasites are divergent unicellular eukaryotes and still cause the death of over 600.000 people per year [1]. To proliferate in the red blood cells of their human host, they use an unconventional cell division mode called schizogony [2–6]. It involves multiple asynchronous rounds of closed mitosis followed by a final round of parasite budding, releasing 12–30 daughter cells from the bursting host cell [7] (Fig 1A). The high number of progeny promotes rapid proliferation and is therefore directly linked to disease severity [8]. Nuclear multiplication requires formation and duplication of the parasite microtubule organizing center (MTOC), also called centriolar plaque, which substantially differs from the highly structured spindle pole body of yeast and the centriole-containing mammalian centrosome [9,10]. Centriolar plaques consist of an amorphous chromatin-free intranuclear compartment from where all mitotic spindles originate, which connects through the nuclear envelope to a protein dense extranuclear compartment, where centrins localize [11–15]. Centrins have various proposed functions and are one of the most widely conserved component of eukaryotic MTOCs [9,16–18]. In yeast, they localize to the half-bridge of spindle pole bodies, where they are required for duplication [19]. In mammalian cells, centrins are found inside the centrioles and are implicated in the function of centrosomes [20–23]. How such a small and well-conserved structural protein functions within those divergent sub-cellular contexts is still unclear. Centrins contain four EF-hand (EFh) domains that can chelate calcium with high affinity, causing conformational changes [24,25]. Calcium-dependent self-interaction of centrins has been documented in a range of eukaryotes including for human centrin 2 (HsCen2) [24,26–29], although the nature of this interaction is unknown. While yeasts encode one centrin, humans have three centrins but only HsCen2 and 3 are directly associated with the centrosome in all tissues [21,30]. The centrin protein family in *Plasmodium* spp. has expanded to four members (Fig 1B), of which three are essential in the blood stage of infection [13,14,31,32]. While the C-terminal sequence containing the EFh domains is homologous, their N-terminus is more variable and can contain phosphorylation sites [33–36]. Centrin relocalization from the cytoplasm to the centriolar plaque at the onset of schizogony has been suggested [12–14]. Which mechanisms drive centrin accumulation in the context of divergent eukaryotic centrosomes is unclear. Here we use biochemical assays with recombinant centrins to reveal their assembly by calcium-dependent liquid-liquid phase separation in vitro. A combination of live cell and super resolution imaging with a novel inducible overexpression system in *P. falciparum* blood stage parasites reveals the conditions required for centrin condensate formation in vivo.

## Results

### All *P. falciparum* centrins localize to the centriolar plaque where they undergo dynamic rearrangement

To assess the localization of PfCen1-4, we episomally expressed GFP-tagged proteins (S1 Fig) in cultured *P. falciparum* blood stage parasites. Previous studies have suggested that *P. berghei* centrins1-3 are refractory to endogenous tagging at the C- and N-terminus [13] and our

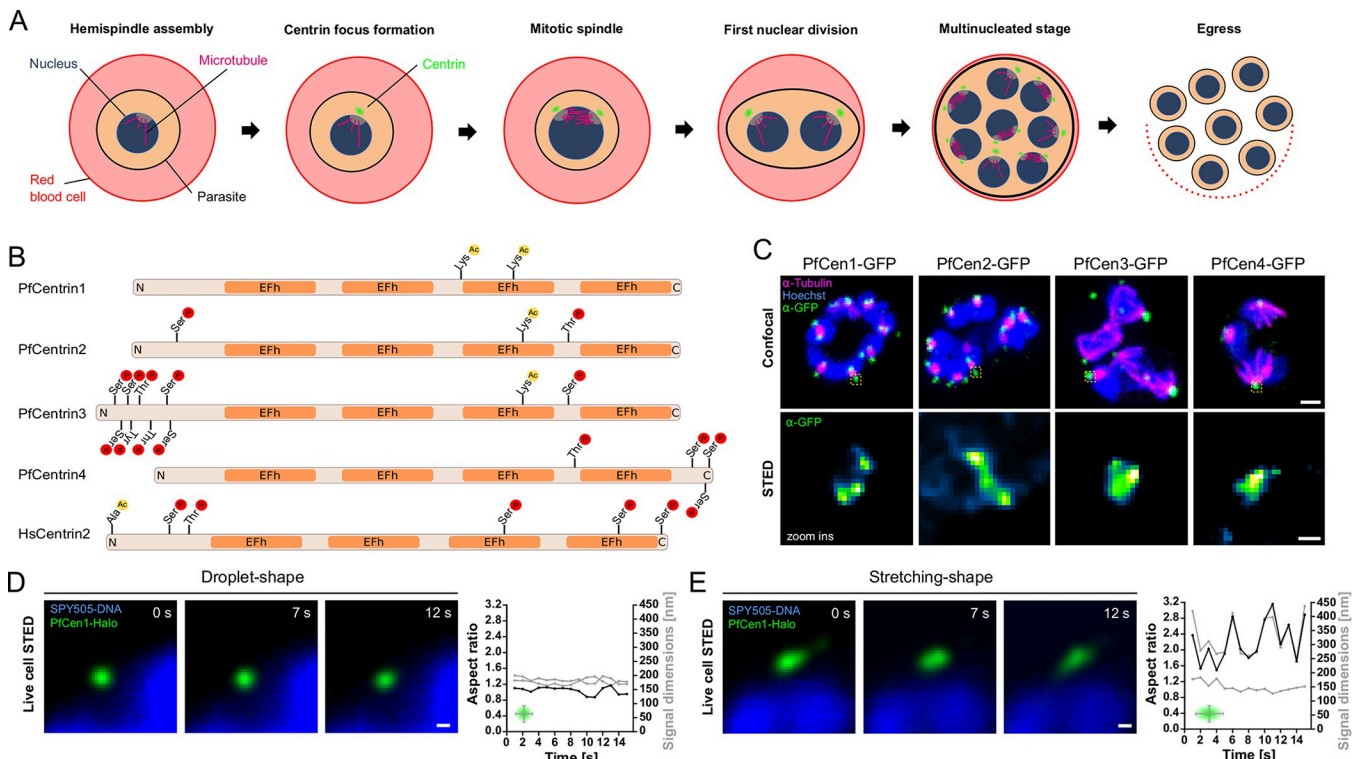

**Fig 1. PfCen1-4 localize to centriolar plaque and can display liquid-like dynamics.** (A) Schematic of nuclear and cell division during asexual blood stage schizogony (B) Schematic of PfCen1-4 and HsCen2 indicating reported post-translational modifications and EFh domains. (C) Immunofluorescence staining of tubulin and PfCen1-4-GFP in parasite strains. All images are maximum intensity projections (MIP). DNA stained with Hoechst. Scale bars; confocal, 1 μm, STED, 100 nm. (D-E) STED time lapse of centriolar plaque region of parasites expressing PfCen1-Halo labeled with MaP-SiR-Halo dye. DNA stained with SPY505-DNA. Scale bar, 100 nm. Quantification of ratio (black line) between height and width (grey lines) of the PfCen1-Halo signal, as indicated in the small schematic.

experience confirms that tagging of the C-terminus of PfCen1 and 3 is not achievable even using small tags (see also Materials and Methods). Episomally expressed PfCen1-4-GFP localized to the centriolar plaque during schizogony (Fig 1C), and Stimulated Emission Depletion (STED) microscopy showed heterogenous shapes of the fluorescent signal. To address whether this heterogeneity might be the result of a dynamic reorganization of centrin within this diffraction-limited region, we implemented live cell STED using a parasite line expressing PfCen1 tagged with Halo (S1 Fig) and labeled with the MaP-SiR-Halo fluorogenic dye [37] (Fig 1D–1E). This either revealed 'wobbling' droplet-like shapes (Fig 1D, S1 Movie), or structures that stretched and retracted along a defined axis (Fig 1E, S2 Movie). These dynamics were reminiscent of biomolecular condensates [38], concentrated protein droplets that form by liquid-liquid phase separation (LLPS), which has emerged as a biological principle explaining the formation of membrane-less organelles, including centrosomes [39].

### *P. falciparum* and human centrins can undergo calcium-dependent liquid-liquid phase separation

To test whether centrins can phase-separate, we expressed 6His-tagged PfCen1-4 and HsCen2 in *E. coli* (S2 Fig) and imaged concentrated recombinant protein solutions. After addition of calcium, we observed rapid droplet formation for PfCen1, PfCen3, and HsCen2 (Fig 2A, S3–S5 Movies). The droplets fused and showed surface wetting, both hallmarks of LLPS [40].

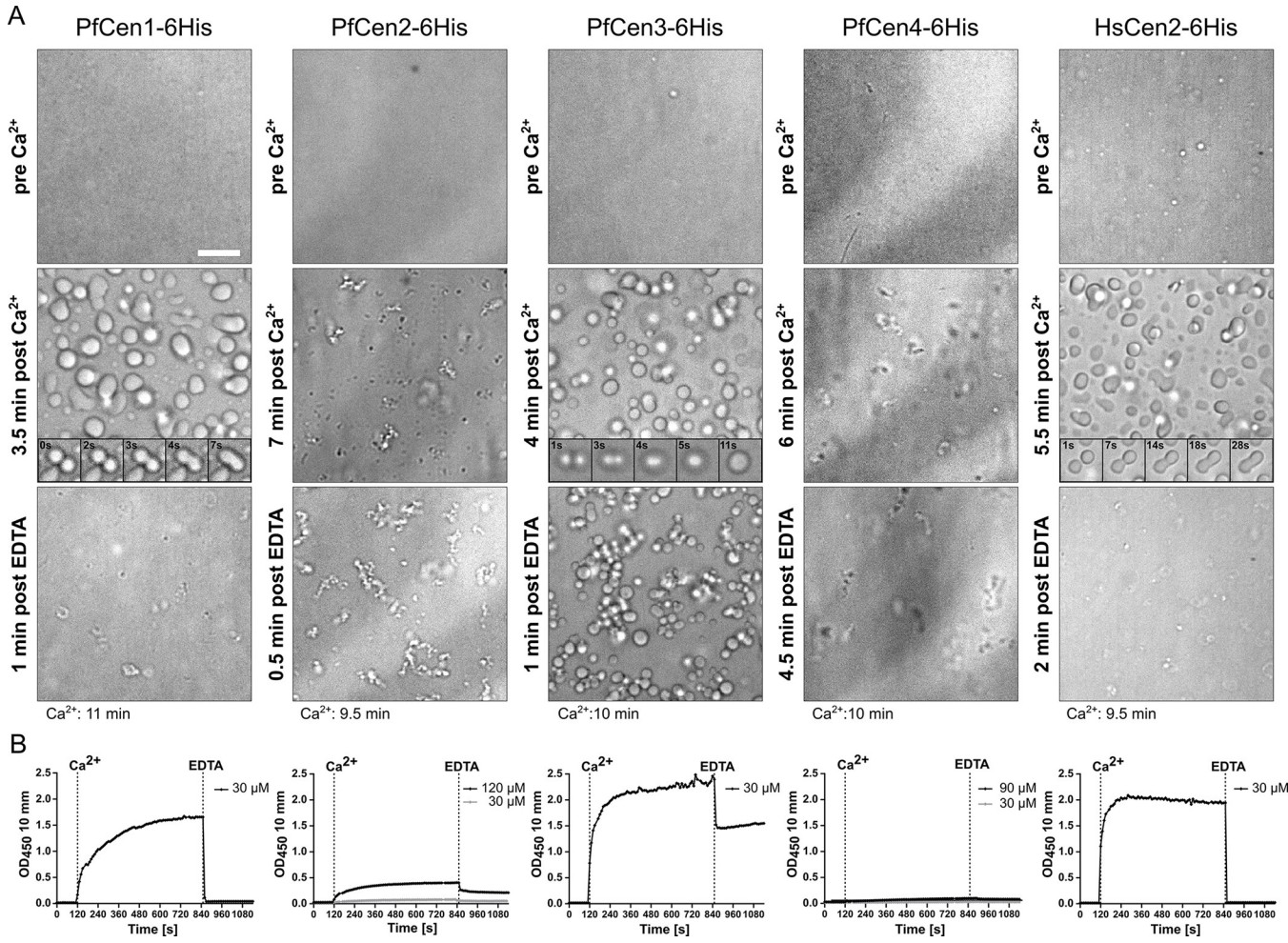

**Fig 2. PfCen1, 3 and HsCen2 undergo calcium-dependent and reversible LLPS in vitro.** (A) Widefield image of highly concentrated recombinant centrin solutions to promote high droplet density of PfCen1-6His (200 μM), PfCen2-6His (120 μM), PfCen3-6His (193 μM), PfCen4-6His (120 μM), HsCen2 (200 μM) before, after 2 mM CaCl₂ addition, and after 10 mM EDTA addition. Scale bars, 10 μm. Inlays show time lapse images of droplet fusion events. Movie metadata was used to estimate the time after calcium addition (bottom) (B) Turbidity measurements in centrin solutions at 30 μM (and higher concentration where indicated) during calcium addition followed by EDTA. Conditions: 50 mM BisTris (pH 7.1) at 37°C.

Addition of EDTA led to instant dissolution of the droplets, except for PfCen3 where non-coalescing droplets remained (Fig 2A, S6 Movie), which can be interpreted as a condensate maturation process towards a more solid or gel-like state [41]. To quantify centrin phase separation kinetics, we measured the turbidity of the solutions by light scattering (Fig 2B). Upon calcium addition, we observed a rapid increase in turbidity that was caused by protein droplet formation (S3A Fig) in PfCen1, PfCen3, and HsCen2 solutions (Fig 2B). Addition of Mg²⁺ as an alternative bivalent cation did not cause a reaction in PfCen1 and only a very reduced one for PfCen3 and HsCen2, suggesting that the effect was calcium-specific (S3B Fig). By exploring a range of PfCen1 concentrations we identified the critical saturation concentration, $c_{sat}$, i.e., the concentration threshold at which a protein transitions from a one-phase to a two-phase regime, to be around 10–15 μM for the different centrins (S3C Fig), which is within the range of what can be found for in vitro phase separation of other proteins (bio-comp.org.cn/llpsdb) [42]. GFP-tagging of PfCen1 did not disrupt LLPS (S3D Fig), which suggests that this property can be preserved in parasite lines episomally expressing tagged PfCen1 (Fig 1B–1D). To

exclude the possibility that the 6His-tag influences phase separation we repeated our analysis with an untagged version of recombinant centrins (S2C Fig), which revealed that PfCen1 and 3 can still phase separate, while PfCen2 and 4 failed to do so even at very high concentrations (S3E Fig). Since PfCen3 condensation was not completely reversible, we added EDTA at earlier timepoints and found that the irreversible fraction increased over time (S4A Fig). This is compatible with the lack of observable fusion events at later time points after calcium addition (S4 Movie), again suggesting a maturation of the condensate [41]. To further test if PfCen3 assembly occurs via LLPS we imaged droplet formation at a lower temperature, which slowed maturation and we detected more fusion events (S7 Movie). In contrast, maturation in PfCen1 condensates was not observed as they could still be fully disassembled by addition of EDTA 3 hours after induction (S4B Fig). Taken together this demonstrates that centrins have differential capacities to undergo calcium-inducible LLPS.

## Intrinsic disorder in the N-terminal sequence of centrins predicts self-assembly

A common feature of phase-separating proteins are intrinsically disordered regions (IDRs) [38], which we analyzed for the different centrins. Indeed, the prediction tool IUPred3 showed high IDR probabilities within the N-termini of PfCen1, PfCen3, and HsCen2, but not for PfCen2 and PfCen4 (S5A Fig). Hence, presence of a N-terminal IDR predicts LLPS for centrins in the tested cases. Analysis for IDRs using IUPred3 in centrins from a range of eukaryotes suggests that most of those species encode at least one IDR-containing centrin (S5B Fig). For some centrins from distant eukaryotes, 'polymer-like' behavior was previously noted in vitro, suggesting that LLPS might be an evolutionary conserved feature of a subset of centrins [26–28]. To further test this hypothesis, we selected IDR-containing centrins from three evolutionarily distant unicellular eukaryotes i.e., *Trypanosoma brucei* (TbCenA), *Chlamydomonas reinhardtii* (CrCen) and *Saccharomyces cerevisiae* (ScCdc31) for recombinant expression (S2C Fig). They showed increases in turbidity upon calcium addition, although Cdc31 required a higher protein concentration (Fig 3A). This may be related to the comparatively lower predicted intrinsic disorder of its protein sequence (Fig 3B). Microscopic analysis confirmed that in vitro all three centrins form fusogenic droplets that display surface wetting and dissolve upon EDTA addition (Fig 3C, S8–S10 Movies), which indicates that LLPS likely is a conserved feature for multiple centrins.

## Phase-separating centrins can form co-condensates in vitro

Interactions between centrins in malaria parasites have already been demonstrated [13,43] and are supported by our localization data (Fig 1C). To test whether different centrins could phase separate together, we determined the saturation concentration $c_{sat}$ for PfCen1 and 3. Under the conditions presented here, we found $c_{sat}$ to be around 10 μM for both and detected no LLPS even when increasing molecular crowding by adding 20 μM of BSA (Fig 4A and 4B). When, however, replacing BSA with the other centrin, we observed clear LLPS at the same total protein concentration indicating that PfCen1 and 3 promote each other's condensation (Fig 4C). This effect is likely due to co-condensation into joint PfCen1-PfCen3 droplets, as shown by incorporation of PfCen1-GFP into preformed PfCen3 droplets (Fig 4D). To test whether PfCen2 and PfCen4 might synergistically contribute to centrin co-condensation we measured turbidity for several combinations (S6 Fig). Consistent with their inability to form condensates on their own PfCen2 did not further contribute to condensation while PfCen4 might have a slight inhibitory effect.

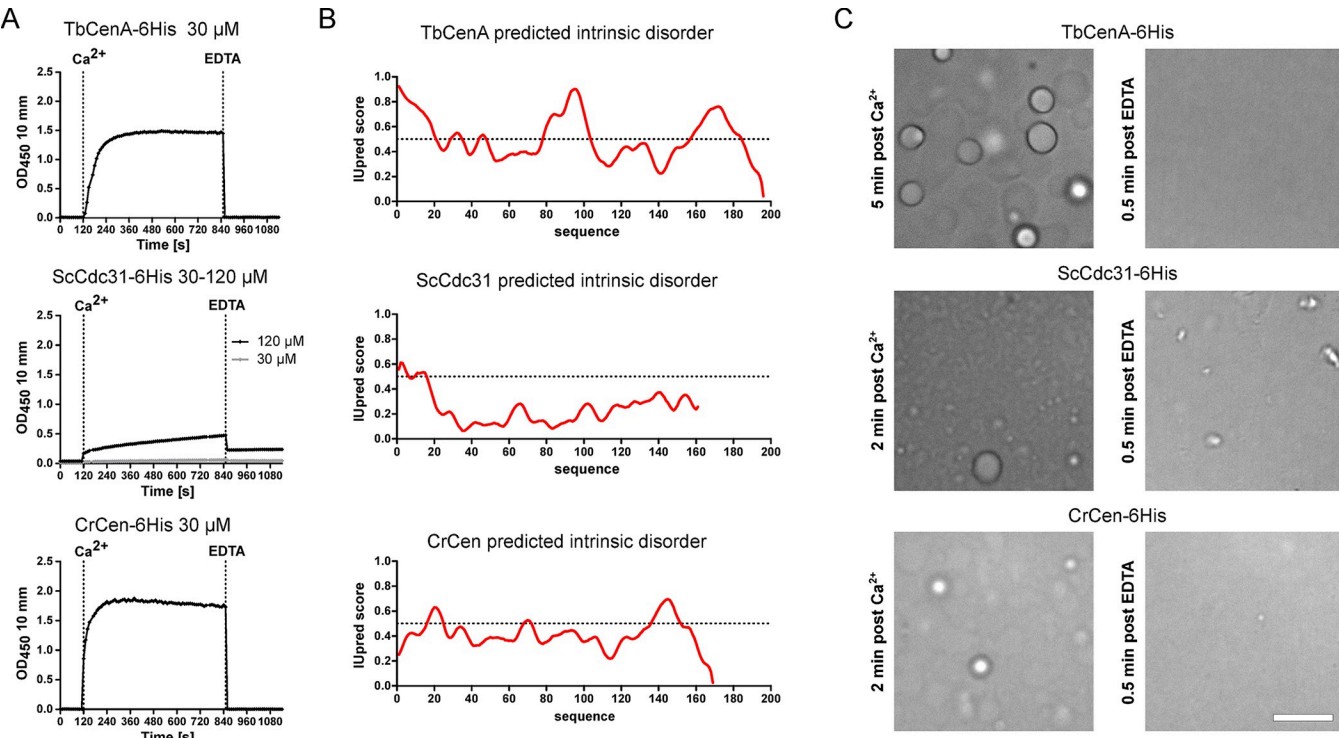

**Fig 3. Centrins from highly divergent eukaryotes can undergo phase separation in vitro.** (A) Turbidity measurements of recombinant *T. brucei* centrin A (TbCenA, 196 aa), *C. reinhardtii* centrin (CrCen, 169 aa) and *S. cerevisiae* centrin (Cdc31, 161 aa) at 30 μM (and higher concentrations where indicated) during calcium addition to 2 mM followed by 10 mM EDTA. Conditions: 50 mM BisTris (pH 7.1) at 37°C. (B) Using the IUpred 3.0 prediction we find a correlation between intrinsic disorder region probability (red line) and LLPS in vitro. (C) Widefield images of recombinant centrin solutions of TbCenA (100 μM), ScCdc31 (200 μM), CrCen (100 μM) after 2 mM CaCl$_2$ addition show droplets and surface wetting, which dissolves quickly after addition of 10 mM EDTA. Scale bar, 10 μm.

## Centrin overexpression causes premature formation of in vivo assemblies with condensate-like properties

While live cell STED microscopy hinted at liquid-like dynamics (Fig 1D–1E), we next investigated whether condensation of centrins can occur in vivo by overexpressing centrins. Since phase separation is concentration-dependent, we speculated whether this would induce additional formation of condensates in parasites. To exert better control over the expression levels

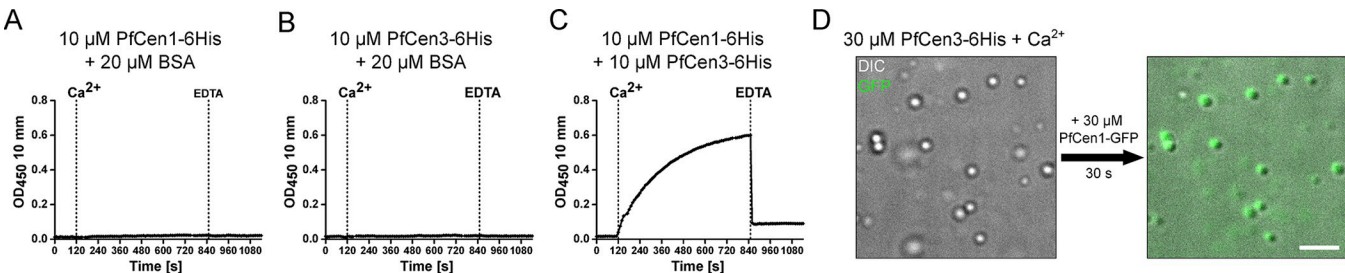

**Fig 4. PfCen1 and PfCen3 interact through co-condensation.** (A-C) Turbidity measurements of recombinant PfCen1-6His and PfCen3-6His at individually subcritical concentrations either independently or as a mixture during addition of calcium followed by EDTA. Individual centrins were supplemented with BSA to a consistent total protein concentration of >20 μM. (D) Brightfield and fluorescence imaging of preformed 30 μM PfCen3-6His protein droplets in presence of Ca$^{2+}$ before and 30 s after addition of recombinant PfCen1-GFP-6His at 30 μM. Focus was adjusted between timepoints. Scale bar, 10 μm. All conditions: 50 mM BisTris (pH 7.1) at 37°C.

of PfCen1-GFP than with the classical overexpression vector (S1 and S7A Figs), we designed a novel *P. Falciparum* Inducible Overexpression (pFIO) plasmid to be transfected in a DiCre recombinase expressing acceptor strain (Fig 5A). Upon addition of rapamycin the dimerized recombinase excises the first open reading frame, placing the gene of interest immediately downstream of the promoter. We designed a version with the medium strength *hsp70* promoter fragment (pFIO) and the stronger *hsp86* promoter (pFIO+), for which we confirmed a substantial increase in median expression levels upon induction (S7B Fig), while some parasites remained GFP-negative. To estimate the degree of overexpression achieved by pFIO +-Cen1-GFP we quantified the ratio between PfCen1-GFP and endogenous PfCen1 at around 18:1 by Western blot using an anti-PfCen1 antibody, while correcting for the fraction of GFP-negative cells (S8 Fig). In cells displaying those high overexpression levels, we frequently observed accumulations of PfCen1-GFP signal that were not associated with any spindle structure or the nucleus. Those Extra-Centrosomal Centrin Accumulations (ECCAs) occurred in 56% (n = 54) of cells carrying the medium strength promoter (Fig 5B) and in 97% (n = 60) of cells with the strong promoter (Fig 5C). ECCAs are not observed in wild type cells when labeling endogenous centrin [12,44]. When quantifying PfCen1-GFP fluorescence intensity within a cell population carrying the same plasmid the number of ECCAs correlated positively with total cell fluorescence intensity, despite the reduced range of observed signal intensities (S9 Fig). When reevaluating our earlier data from the weaker expressing pARL-Cen1/3-GFP parasite lines, we could occasionally detect ECCAs, although their frequency was too low for systematic quantification. When measuring the dimensions of the centrin signal at the centriolar plaque we found that upon pFIO+-Cen1-GFP overexpression the width (321 vs 324 nm) was unchanged, while the height (219 vs 247 nm) showed an increase of 11% (S10 Fig). To test if overexpression of PfCen1-GFP affects the dynamics of centrin assembly, we used time-lapse microscopy. Initially, cells showed a homogenous cytoplasmic distribution, while ECCAs frequently assembled before the formation of the first mitotic spindle (Fig 5D, S11 Movie). Centrin accumulation by biomolecular condensation predicts that once a critical concentration threshold is crossed, additional centrin would only accrue into the condensate fraction. Machine-learning-based image analysis showed that PfCen1-GFP concentration in the cytoplasm remained stable, while the fraction of PfCen1-GFP in foci increased once their formation started (Fig 5E). The partition coefficient of centrin signal between cytoplasmic and foci fraction was stable around 4.0 until it slightly increased to 5.3 after mitotic spindle formation (S11 Fig). At the end of schizogony, centrin foci disappeared, while the total centrin signal stayed constant, which indicates that centrin assemblies are dissolved (Fig 5E). Usually centrin localizes to the centriolar plaque at the onset of schizogony as marked by the formation of the first mitotic spindle [12]. To further test if overexpression of PfCen1-GFP affects the dynamics of centrin assembly, we quantified the first appearance of ECCAs relative to mitotic spindle formation (Fig 5F). Indeed, PfCen1-GFP coalesces into ECCAs and at the centriolar plaque prematurely but not earlier than a few hours before the entry into the schizont stage. The timing of both ECCA and centriolar plaque foci formation was, however, concentration-dependent, which can best be explained reaching the critical concentration for schizogony-associated condensation earlier [12]. ECCAs often formed first at the centriolar plaque before detaching, which further suggests an involvement of schizogony-specific nucleation factors (S12 Movie). To test if ECCAs and centrosomal centrin foci contain protein aggregates, we stained PfCen1 overexpressing parasites with Proteostat (Fig 5G) [45]. Both structures were negative, suggesting that they are not a mere product of protein aggregation.

To exclude that ECCAs only form due to GFP-tagging we designed a version of pFIO + where GFP and PfCen1 were separated by a T2A skip peptide (Fig 5H). During translation the T2A sequence causes the ribosome to "skip" bond formation between amino acids and

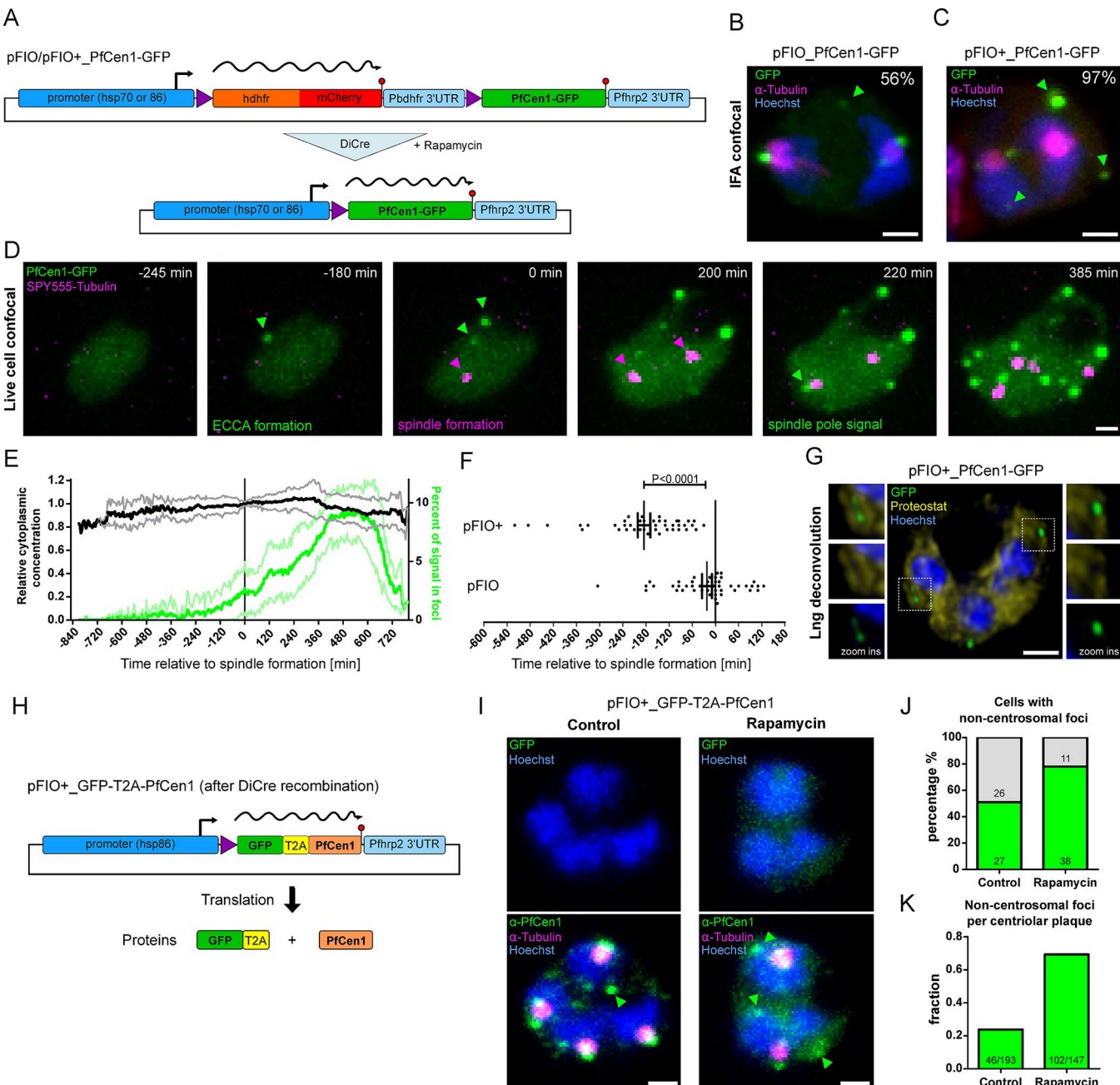

**Fig 5. PfCen1 displays condensate-like properties in parasites.** (A) Schematic of pFIO plasmids during DiCre-dependent recombination. (B) Immunofluorescence of tubulin and GFP in parasites moderately overexpressing PfCen1-GFP (pFIO). Percentage of early schizonts containing ECCAs (arrows) indicated. (C) as in (B) but for cell strongly overexpressing PfCen1-GFP (pFIO+) (D) Time lapse images of parasites overexpressing PfCen1-GFP from pFIO+ and labeled with microtubule live stain SPY555-Tubulin (E) Normalized mean cytoplasmic PfCen1-GFP fluorescence intensity over time and share of PfCen1-GFP fluorescence signal contained within foci with standard deviation. n > 33. (F) Quantification of timepoint of PfCen1-GFP foci appearance relative to first mitotic spindle formation detected by SPY650-tubulin comparing pFIO/pFIO+; standard error of the mean, two-tailed t-test, $n_1 = 46$, $n_2 = 45$. (G) Proteostat and PfCen1-GFP live cell staining. (H) Schematic of pFIO+ version containing T2A skip peptide and expected translation products. (I) Immunofluorescence of rapamycin-induced and control parasites carrying pFIO+_GFP-T2A-PfCen1 plasmid labelled with anti-PfCen1 and anti-Tubulin antibodies. Arrow heads indicate foci above threshold. All images are MIP. DNA stained with Hoechst. Scale bars; 1 μm. (J) Quantification of cells having non-centrosomal foci (which includes unspecific background dots and ECCAs) in induced (n = 49) and control (n = 53) conditions. Absolute numbers given in columns. (K) Quantification of number of non-centrosomal foci per centriolar plaque.

produces two separated proteins from the same mRNA [46]. In this parasite line, PfCen1 only contains a single additional N-terminal proline and Western blot analysis of induced parasite lysate confirmed efficient "skipping" (S12 Fig). To visualize the localization of untagged PfCen1 we produced a rat polyclonal anti-PfCen1 antibody, which detected a specific band in parasite protein extract albeit showing some background signal and cross-reactivity with the highly homologous PfCen3 (S13 Fig). Using this antibody, we immunolabeled the rapamycin-induced cells carrying pFIO+_GFP-T2A-PfCen1 and specifically selected GFP-positive ones in the population for imaging (Fig 5I, top panels). Due to unspecific labeling (S13 Fig), the background of the PfCen1 antibody was higher than for the anti-PfCen3 and produced additional unspecific cytoplasmic dots even in non-overexpressing cells (Fig 5I, bottom panels). Therefore, we set an intensity threshold based on centrosome foci intensity to reduce the impact of unspecific signals in the non-induced conditions. We consistently observed more non-centrosomal anti-PfCen1 foci in induced cells and found a higher number of cells carrying non-centrosomal foci in induced parasites, which likely present unspecific dots and ECCAs together (Fig 5J). When looking at the number of non-centrosomal foci detected per centriolar plaque in all cells the effect of PfCen1 overexpression was even more clearly indicating that ECCAs are formed, and their presence does not depend on GFP-tagging (Fig 5K). These data are concordant with a regulated and concentration-dependent condensation mechanism of PfCen1 in vivo.

## Hexanediol treatment causes centriolar plaque disassembly

Biomolecular condensates are frequently formed by weak non-polar interactions, that can under some circumstances be dissolved by 1,6-hexanediol. Addition of 5% 1,6-hexanediol to phase-separated PfCen1 in vitro only had a moderate effect on droplet dissolution (Fig 6A). This is compatible with previous observations that centrin assembly also has a strong polar component [26,27]. To probe this in a physiological context we also treated wild type parasites for 5 min with 5% 1,6-hexanediol followed by anti-PfCen1 immunolabeling (Fig 6B). We observed a striking dissolution of centrin foci and more than 74% of spindles lost centrin foci at the centriolar plaque compared to 2% in control cells (Fig 6C), suggesting that non-polar interactions might play a more important role for centrins in vivo or that the centriolar plaque integrity as a whole might be affected. Taken together these observations argue for condensation being part of the mechanism of centrin assembly at the centriolar plaque.

## PfCen1 requires its N-terminus and calcium-binding activity for condensation

To obtain insights into the molecular determinants of centrin condensation, we set out to functionally interfere with LLPS in vitro and in vivo. When deleting the IDR-containing N-

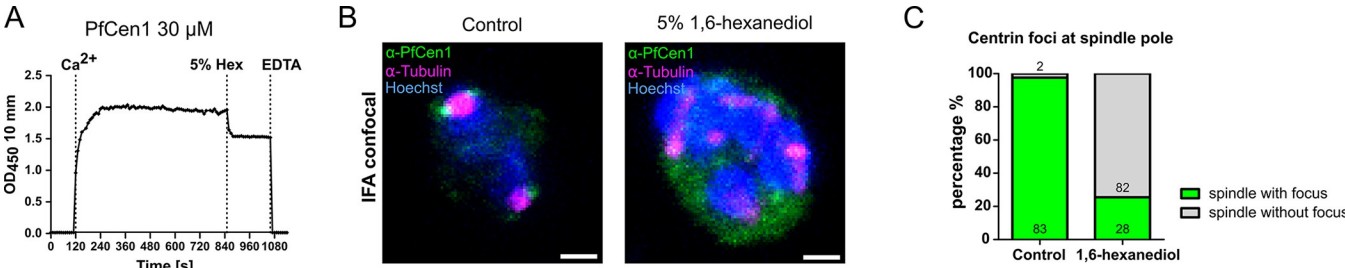

**Fig 6. 1,6-hexanediol treatment dissolves centrin foci in vivo.** (A) Turbidity measurements of recombinant PfCen1 solution at 30 μM in 50 mM BisTris (pH 7.1) at 37°C during addition of calcium followed by 5% 1,6-hexanediol and EDTA. (B) Immunofluorescence of non-overexpressing parasites treated with 5% 1,6-hexanediol or untreated control for 5 min stained with anti-PfCen1 and anti-Tubulin antibodies. Images are MIP. DNA stained with Hoechst. Scale bars; 1 μm. (C) Quantification of spindles in treated (n = 110) and control (n = 85) cells (early schizonts) with or without centrin focus at the spindle pole in percentage. Absolute numbers given in columns.

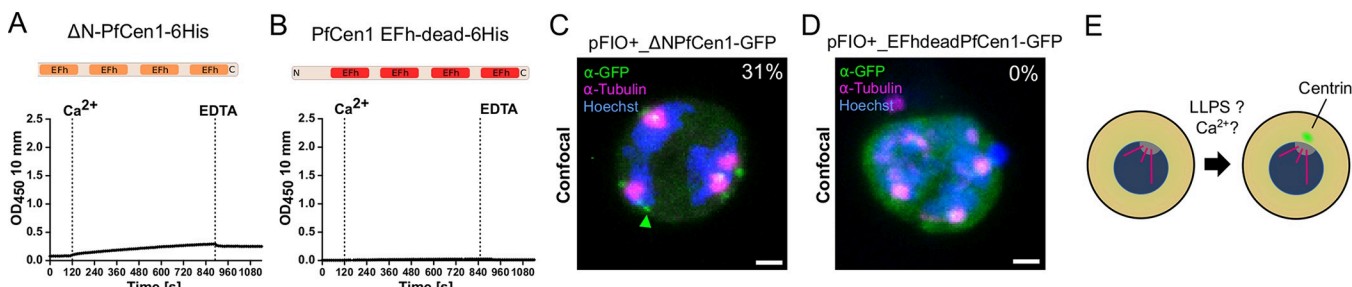

**Fig 7. PfCen1 phase separation depends on its N-terminus and calcium binding.** (A-B) Turbidity measurements in recombinant protein solutions at 30 μM of PfCen1 mutants lacking the N-terminus or with disabled EFh domains, respectively, during addition of calcium followed by EDTA. Conditions: 50 mM BisTris (pH 7.1) at 37˚C. (C-D) Immunofluorescence staining of tubulin and GFP in parasites overexpressing N-terminal deletion and non-$Ca^{2+}$-binding mutant of PfCen1-GFP from pFIO+. Percentage of cells containing ECCAs (arrows) indicated. All images are MIP. DNA stained with Hoechst. Scale bars; 1 μm. (E) Our findings suggest that centrin accumulation at the centriolar plaque from a pre-mitotic diffuse cytoplasmic pool depends on LLPS, which might be promoted by cellular signaling events like calcium levels and a nucleating factor at the centrosome. Parasite is shown without host red blood cell for clarity.

terminus from recombinant PfCen1 (S2C Fig), we observed strongly reduced LLPS and merely detected aggregation (Fig 7A). LLPS was not rescued by replacement with the IDR-free N-terminus of PfCen4, neither was the N-terminus of PfCen1 sufficient to confer LLPS to a chimeric PfCen4 (S14 Fig). To test the role of EFh domains, we generated a EFh-dead mutant by introducing four critical D to A mutations in the $Ca^{2+}$-binding pocket, which was shown to strongly inhibit $Ca^{2+}$-binding (Fig 2C) [47]. Circular dichroism spectroscopy showed that the secondary structure of PfCen1 was not affected by those four point mutations (S15 Fig), yet in this mutant, LLPS was abolished (Fig 7B).To test the significance of the IDR in vivo, we overexpressed the N-terminal deletion mutant, observing ECCAs in only 31% (n = 55) of cells (Fig 7C), as compared to 97% upon pFIO+ overexpression of PfCen1 (Fig 5C). The EFh-dead PfCen1 mutant (Fig 7D), lacked any discernible formation of foci in vivo, suggesting a critical role for Ca2+ responsiveness for targeting and condensation. Together this suggests that active EFh-domains and a disordered N-terminus are essential but not sufficient for calcium-induced phase separation of centrins, with alterations inhibiting LLPS in vitro also affecting foci formation in vivo. However, deletion of the N-terminus retains some targeting activity while EFh-dead PfCen1-GFP stays completely soluble.

## Discussion

### Working model for centrin accumulation

Centrins are conserved throughout eukaryotes and implicated in a wide range of processes, which require their assembly [24]. Our study in an early branching eukaryote reveals calcium-inducible phase separation as a new centrin assembly principle (Fig 7E). Centrins frequently associate with highly structured and polymeric cellular complexes such as the base of flagella and cilia, the inside of centrioles, the basal body and the associated contractile fibers [23,48–50]. Yet there is little evidence that centrins themselves form polymers. An early study posited that a centrin of the algae *Scherffelia dubia* forms a filamentous network visualized by electron microscopy albeit using a protein concentration of 1500 μM [51]. These excessively high concentrations can result in spinodal decomposition, a type of transition into a two-phase regime, where thermodynamic barriers to phase separation disappear and proteins can form network like structures [52]. The diameters of the observed network branches were further too thick to correspond to filamentous arrangement of centrin monomers [51]. An early physicochemical analysis of calcium dependent self-assembly of HsCen2, however, suggested the existence of "more complex energetics of the polymer solution"[26]. The same study showed dependencies

of centrin self-assembly on concentration, pH, and temperature all in line with an expected effect on LLPS. While polymerization of centrins cannot be definitively ruled out, our study suggest that LLPS is relevant for their assembly in several cases.

## Centrin dynamics and overexpression in human cells

While centrins have been associated with a range of functions including nuclear export, transcription, and DNA excision repair, their association with MTOCs plays a dominant role [24,48,53,54]. We support this notion by confirming that all four centrins are focused at the centriolar plaque in *P. falciparum*, while other potential localizations are below our detection threshold. MTOCs can be viewed as membrane-less organelles and biomolecular condensation has been ascribed to various centrosomal proteins [39]. Here, the model posits that phase-separating proteins, which can act as scaffold, concentrate so called client proteins [55]. An example is the enrichment of tubulin monomers by SPD-5 to assist and control their polymerization into microtubules [41]. It is, however, important to note that in asexual blood stage parasites the centrin-containing compartment and the spindle microtubule nucleation sites are physically separated by the nuclear membrane [12], while in male gametocytes centriole-like structures that will act as basal body for the growing axonemes are formed in direct proximity to centrin [15]. A study using laser ablation of human centrosomes to study de novo centriole formation showed that it begins with the "coalescence" of small centrin "aggregates" that appear amorphous in EM [56]. In S-phase arrested cells formation of "centrin granules" was observed at the nuclear periphery [57]. HsCen1-GFP overexpression further accentuates formation of nucleus-associated and coalescing centrin granules and causes centrosome overduplication. Another study overexpressing GFP-tagged HsCen2 also found an excess of centrin foci, which caused centriole overproduction, and this effect was reproducible without tag [58]. These structures as well as centriolar satellites in general are reminiscent to the ECCAs we observe in the malaria parasite. The observations that Cdk2 activity was required for centrin foci formation and that the overexpression phenotypes varied between human cell types and stages suggests that foci assembly is regulated [57–59].

## Role of calcium in centrin assembly

In line with regulated assembly, we observed centrin accumulation, even in overexpressing parasites, only at the entry into the nuclear division stage, while the non-calcium-binding EFh-dead PfCen1 mutant failed to accumulate at all. This calcium dependency of centrin LLPS provides a possible link to calcium signaling, which plays multiple roles [60]. The reported increase in intracellular calcium associated with schizogony could therefore help to promote timely centrin accumulation at the centriolar plaque [61]. Interestingly, human cells display a functional calcium spike at the centrosome during mitosis [62]. Our attempts to reduce intracellular calcium concentrations to impact centrin dynamics in vivo were unfortunately inconclusive. A central caveat is that effective removal of intra-parasitic calcium likely requires lysis of the host red blood cell [61,63,64]. Yet, it is unlikely that centrin accumulation at the centriolar plaque is solely regulated by calcium but rather by various cell cycle regulation pathways, as has been described for human centrins [30,58,59]. Cell cycle regulation could for example influence the phosphorylation status of the N-terminus of PfCen3 (Fig 1B).

## Predicting centrin LLPS in vivo

To demonstrate the significance of LLPS in vivo is one of the more challenging aspects of the field, even in model organisms [52,65,66]. As one of the defining features of phase separation the question whether the saturation concentration, $c_{sat}$, for a given protein can be reached in a

cell is crucial. In vitro studies can give clues about $c_{sat}$, but solution conditions significantly affect protein behavior [40]. Addition of crowding agents like PEG and decreasing temperature can significantly reduce $c_{sat}$, while addition of salt can increase it. For PfCen1 we found a $c_{sat}$ of about 10–15 μM at 37° C while adding neither salt nor PEG. This lies within the range of what has been quantified for other LLPS proteins [42]. In the much more complex environment of a cell the saturation of the dilute phase is determined by a convolution of macromolecular concentrations, and therefore vastly differs from the in vitro conditions, which makes a prediction of $c_{sat}$ in vivo challenging [65]. Our observation that PfCen1 and 3 form co-condensates would also require that the joint concentration of both proteins must be known. Measuring intracellular protein concentrations in *P. falciparum* is challenging, particularly when the endogenous protein, such as is the case with PfCen1-3 cannot be tagged. Here, overexpression offers an opportunity to assess whether phase separation-like behavior of centrins in cells is possible at all and which are the necessary structural features of the protein to form condensate-like structures. Whether centrin overexpression affects parasite fitness remains to be clarified.

## Principles affecting saturation concentration

In PfCen1-GFP overexpressing schizonts just before centrin foci appearance (Fig 4D–4F) the cytoplasmic centrin concentration appears to be below the critical threshold. Nevertheless, mechanisms such as surface-assisted condensation and prewetting have been described to promote condensation below the cytoplasmic $c_{sat}$ [67–71]. Examples ranging from bacteria to human cells have been observed where a polymer enriches a condensing protein [72,73]. In bacteria, e.g., filamentous PomX recruits PomY condensates by surface-assisted condensation, which in turn enrich and bundle bacterial tubulin-like FtsZ polymers. Prewetting can be envisioned as wetting of dewdrops on a spider web and has been described for transcription factors condensing on isolated DNA strands [68]. We previously described a novel Sfi1-like protein (PfSlp) interacting with centrins at the centriolar plaque [43]. It is characterized by a large size, likely taking an elongated shape, and the presence of multiple centrin binding motifs along its sequence, which is itself highly disordered. This offers the potential for multivalent interaction, which can dramatically decrease $c_{sat}$ [52]. Whether PfSlp forms antiparallel bundles like yeast Sfi1 along which centrins could form condensates is speculative [74,75], but would be coherent with the dimensions of the stretched PfCen1-Halo signal tangentially to the nucleus (Fig 1E). Deformation of a condensate from a round shape can indeed be achieved through external forces and cytoskeletal elements [70]. The observation that ECCAs are initially formed at the centriolar plaque and therefore the nuclear membrane surface is compatible with a surface-assisted condensation or prewetting transitions. This remains of course highly speculative and it must be investigated further whether non-homotypic interaction, which can either promote or antagonize phase separation, could be the dominant cause of the observations made [66].

## Biomolecular condensation manifests on a spectrum

While the ability of individual proteins to phase separate is undisputed by now, the debate whether biomolecular condensation is the dominant principle behind formation of membrane-less organelles, such as the centrosome, is still ongoing [76]. Some of this debate can be attributed to a differential use of nomenclature as biomolecular condensate formation does e.g. not per se imply phase separation [65]. As the field progresses it becomes clear that biomolecular condensation unfolds over a range of interaction types and biophysical properties of proteins and nucleic acids, which has led to proposing the term PS++ for all associated phenomena [65]. Where centrins are positioned within this spectrum and how variable the biophysical properties of centrin condensates are shall be investigated in the future [77].

## Potential implications for centriolar plaque biology

What remains is the question whether phase separation is an adequate conceptual framework to understand the functional questions surrounding centriolar plaque biology. What our study shows is that centrins only accumulate in vivo when the parasite is approaching its nuclear division stage and phase separation is not required for all centrins, i.e., PfCen2 and 4, to be recruited to the centriolar plaque. Hence there must be mechanisms that promote timely assembly, but also dissolution, of centrin foci throughout the course of schizogony. As in many cases identifying genuine biologically importance for centrin phase separation remains limited [52]. However, envisioning phase-separating scaffold proteins and recruited clients is useful to conceptualize assembly of the amorphous centriolar plaque [55,78]. Phase separation could help explain how the centriolar plaque remains fluid enough to allow splitting during subsequent nuclear division cycles, while limiting the number of newly forming assemblies [12,15]. Despite the caveats of 1,6-hexanediol treatment [79], the ensuing dissolution of centrin foci are a first indication that biomolecular condensation might play a role in *P. falciparum* MTOC biology.

# Materials and methods

## Cloning of plasmids

All plasmids were assembled via Gibson assembly using the HiFi DNA Assembly mix (New England Biolabs), unless stated otherwise. Correct sequences were confirmed via Sanger sequencing (Eurofins Genomics, SupremeRun). PCRs were performed using Phusion polymerase (New England Biolabs). Details on primer sequences, templates, and synthetic oligos or genes are provided in S1 and S2 Tables. Plasmids were amplified in chemically competent 5-alpha F'Iq *Escherichia coli* cells (New England Biolabs) after 42˚C heat shock transformation and extracted using the GenElute HP Plasmid Miniprep Kit (Sigma-Aldrich, Merck). To generate pARL-PfCen1-GFP, -PfCen2-GFP, and PfCen4-GFP from pARL-PfCen3-GFP (kindly provided by Tim Gilberger), the PfCen3 sequence was removed via restriction digest with KpnI-HF and AvrII. It was replaced with PfCen1, PfCen2, or PfCen4 sequences PCR amplified via the primer pairs P1+2, P3+4, or P5+6, respectively. To generate pARL-PfCen1-Halo the GFP sequence was removed from pARL-PfCen1-GFP via restriction digest with AvrII and PstI-HF. It was replaced with the Halo sequence PCR amplified via primer pair P7+P8. To generate the pFIO plasmid pARL-Cen3-GFP was digested with NotI-HF and HindIII-HF, leaving an hrp2 3'UTR. In front of this hrp2 3'UTR a Pfhsp70 5'UTR-LoxP-hDHFR (human dihydrofolate reductase) sequence was inserted from two fragments PCR amplified via primer pairs P9+P10 and P11+12 respectively. This created an intermediate plasmid, which was digested with BamHI-HF (in between the hDHFR and hrp2 3'UTR) to insert an mCherry-LoxP-GFP sequence from two fragments PCR amplified via primer pairs P14+15 and P16+17 respectively. This plasmid was then digested with SalI-HF (in between mCherry and LoxP-GFP) to insert a PbDHFR 3'UTR sequence PCR amplified via primer pair P18+19 to create the basic pFIO_GFP. To generate pFIO+_GFP the promoter was replaced by digestion with NotI-HF and BamHI-HF (which also removed the hDHFR sequence) and integration of Pfhsp86 5'UTR and hDHFR sequences PCR amplified using primer pairs P20+21 and P11+P13 respectively. To generate pFIO/pFIO+_PfCen1-GFP, and pFIO+_ΔN-PfCen1-GFP, previously generated pFIO_GFP and pFIO+_GFP were digested with MluI-HF and XhoI to remove the GFP open reading frame (ORF), which was replaced with a PfCen1-GFP or ΔN-PfCen1-GFP ORF PCR amplified using primer pairs P22+17 or P23+17 respectively. pFIO+_ EFh-dead-PfCen1-GFP, and_HsCen2-GFP were generated by removing the PfCen1 ORF in pFIO+_PfCen1-GFP by digestion with MluI-HF and AvrII, and replacing it with

synthesized genestrings (Thermo Fisher) encoding EFh-dead-PfCen1, where aspartate 37, 73, 110, and 146 were mutated to alanine, and HsCen2 codon optimized for *P. falciparum* (oligo 1 and 2). pFIO+_GFP-T2A-PfCen1, which contained a regular hDHFR gene instead of the hDHFR-mCherry fusion protein, was generated in two steps. First the mCherry sequence was removed from pFIO+_PfCen1-GFP by digestion with BamHI-HF and SalI to insert a ssDNA containing a stop codon (oligo 3) via Gibson assembly. The resulting plasmid was digested with MluI-HF and XhoI removing the PfCen1-GFP ORF and replacing it with a GFP-T2A followed by a PfCen1 sequence PCR amplified using primer pairs P24+P25 and P26+P27 respectively. All plasmodial vectors contained a hDHFR selection cassette conveying resistance to WR99210. pET-28 constructs were generated by digesting pET-28a(+) with NcoI-HF and XhoI and inserting HsCen2 as well as codon optimized PfCen1, ΔN-PfCen1, or PfCen2 sequences PCR amplified using primer pairs P28+29 and P30+31, P32+31, and P34+35 respectively. pET-28_PfCen1-GFP was generated by integrating a PfCen1 and GFP sequence PCR amplified using primer pairs P30+33 and P36+37 respectively. Codon optimized EFh-dead PfCen1, ScCDC31, TbCen2, and CrCen sequences were integrated as synthetic dsDNA oligo 10, oligo 11, oligo 12, and oligo 13 respectively. We generated pZE13d-C, a version of the bacterial expression vector pZE13d with a C-terminal 6His tag. For this purpose, pZE13d was digested with BamHI-HF and PstI-HF. A de novo sequence encoding the 6His tag (followed by BamHI and SalI restriction sites) was formed by annealing oligo 4 and 5 over a 30 min temperature gradient from 95°C to 25°C and integrated via T4 ligase (New England Biolabs). pZE13d-C_PfCen3 and _PfCen4 were generated by digesting pZE13d-C with BamHI-HF and SalI-HF and integrating codon optimized PfCen3 and PfCen4 encoding sequences PCR amplified via primer pairs P38+39 and P40+41. We generated pZE13d-TEV, a version of the bacterial expression vector pZE13d with an N-terminal 6His-Linker-TEV sequence for removal of the 6His-Tag via TEV protease. For this purpose, pZE13d was digested with EcoRV-HF and PstI-HF. The 6His-Linker-TEV encoding sequence (followed by an SfoI restriction site) was integrated as a ssDNA sequence (oligo 14) via Gibson assembly. pZE13d-TEV_PfCen1-4 were generated by digesting pZE13d-TEV via SfoI and integrating PfCen1-4 sequences PCR amplified using primer pair P42+P46, P43+P46, P44+P46, and P45+P46 respectively. All pZE13d derived bacterial vectors contained an ampicillin resistance cassette and all pET-28 derived vectors a kanamycin resistance cassette.

## *P. falciparum* blood culturing

All *P. falciparum* cell lines were maintained in human O+ erythrocyte cultures with a 2.5% haematocrit in RPMI 1640 medium supplemented with 0.5% AlbuMAX II Lipid Rich bovine serum albumin (Thermo Fisher Scientific), 25 mM Hepes (Sigma-Aldrich), 0.2 mM hypoxanthine (c.c.pro GMbH), and 12.5 μg/ml Gentamycin sulfate (Carl Roth) at pH 7.3 (cRPMI). Cultures were incubated at 37°C in 3% $CO_2$, 5% $O_2$, and 90% humidity at 37°C. Parasitemia was maintained below 5% and determined via Hemacolor staining of blood smears (Sigma-Aldrich, Merck) using a Nikon Eclipse E100 microscope (Nikon Corporation) with 100x oil immersion objective. Depending on the application asexual cycle stages were synchronized to the ring stage by incubation of infected red blood cells (iRBCs) in a 25x volume of prewarmed 5% sorbitol (Sigma-Aldrich) solution at 37°C for 10–15 min, followed by a wash step in cRPMI before return to culture.

## *P. falciparum* transfection

For generation of transgenic parasite lines *P. falciparum* NF54 served as the acceptor strain for pARL plasmids and 3D7-DiCre (kindly provided by Anthony Holder and Mortiz Treeck) for

pFIO plasmids [80,81]. Plasmids were purified from 200 ml transformed *E. coli* overnight culture (37˚C, 130 rpm) via the NucleoBond Xtra Midi Kit (Macherey-Nagel) and precipitated by supplementing 0.1x volume 3 M sodium acetate (Sigma-Aldrich, Merck, pH 5.3) and a 2x volume -20˚C ethanol (VWR Chemicals). After overnight incubation at -20˚C the DNA was pelleted at 17000g for 30 min at 4˚C, washed with 70% ethanol, and air-dried under sterile conditions. For transfection 150 µl of iRBCs with a $\geq$ 4% ring stage parasitaemia were washed with prewarmed CytoMix [120 mM KCl, 0.15 M $CaCl_2$, 2 mM EGTA, 5 mM $MgCl_2$, 10 mM $K_2HPO_4$/$KH_2PO_4$, 25 mM Hepes, pH 7.6], then mixed with 100 µg purified plasmids in 30 µl TE buffer [10 mM Tris, 1 mM EDTA, pH 8.0] and 350 µl CytoMix. The mixture was transferred into a pre-cooled (4˚C) Gene Pulser Electroporation cuvette (Bio-Rad) and electroporated using the Gene Pulser II system (Bio-Rad) at 310 kV, 950 µF, exponential mode. iRBCs were then transferred to 5 ml pre-warmed cRPMI for 30 min recovery at 37˚C before centrifugation at 800g for 2 min and transfer to regular culture. Selection for episomal plasmids began 6 h post transfection and was henceforth maintained with 2.5 nM WR99210 (Jacobus Pharmaceutical Company). Tagging of endogenous PfCen1 and 3 with GFP or HA tag using the selection linked integration system were attempted but unsuccessful and yielded no transfectants despite several attempts [82]. Modification of the endogenous PfCen3 gene via CRISPR/Cas9 failed as well.

## pFIO induction

Expression of the second pFIO ORF was induced in synchronous 3D7-DiCre_pFIO/pFIO + parasite lines approximately 48 h prior to experiments during the schizont stage. 25 µl iRBCs were incubated in 200 µl cRPMI supplemented with 100 nM rapamycin (Sigma-Aldrich, Merck) or 1% DMSO (Sigma-Aldrich, Merck) as a control for 4 h at 37˚C. Afterwards cells were pelleted 1 min at 800 g for washing 4 times and cultured in 1 ml cRPMI without selection drug until further processing for imaging or FACS analysis.

## GFP quantification via flowcytometry

For flowcytometric quantification of GFP signal in induced 3D7-DiCre pFIO/pFIO+ strains 2.5 µl iRBCs were washed once with RT 0.9% NaCl solution (B. Braun GmbH) and stained with 5 µM SYTO 61 Red Fluorescent Nucleic Acid stain (Thermo Fisher Scientific) for 1 h at 37˚C in 0.9% NaCl solution and washed once again afterwards at RT. Cells were analysed at RT using the FACSCanto II (BD Biosciences), first gating erythrocytes in the FSC-A/SSC scatter and then single cells in the FSC-H/FSC-A scatter. SYTO 61 signal was acquired using a 640 nm laser and 660/20 emission filter. Of the positive cells, i.e., iRBCs, multinucleated cells, i.e. schizonts, were selected and the share of GFP positive cells determined via a 488 nm laser and 530/30 nm emission filter, for which the gate cut-off was determined based on the uninduced DMSO control.

## *P. falciparum* protein extraction and Western Blot analysis

For analysis of GFP-T2A-PfCen1 T2A skipping efficiency and rat anti-PfCen1 antibody binding, proteins were first extracted from WT NF54 or 3D7-DiCre_pFIO+ GFP-T2A-PfCen1 parasites treated with rapamycin or DMSO 5 days prior. Parasitemia was assessed via blood smear. iRBCs were washed in prewarmed PBS and parasites isolated from RBCs via 0.15% Saponin/PBS. Cells were pelleted at 3300g for 5 min followed by 3 PBS wash steps. Cells were then boiled in 1x Laemmli/PBS buffer to a final concentration of $10^7$ parasites per 10 µl lysate at 95˚C for 10 min. Hemozoin was removed via centrifugation at 21000g for 1 min. 10 µl supernatant parasite lysate or 100 ng recombinant protein were loaded onto 12% Mini-

Protean TGX gels (Bio-Rad). For quantification of PfCen1-GFP to endogenous PfCen1 ratio, 125 μl erythrocytes infected with 3D7-DiCre_pFIO+_PfCen1-GFP at 1% parasitemia were treated with rapamycin or DMSO. 48 h later trophozoites and schizonts were purified using a 67.5% Percoll (Cytiva) gradient at 1500 g for 10 min RT. Parasites were briefly washed in PBS. A subset was then seeded to a 35 mm IbidiTreat dish (Ibidi), fixed for 20 min at 37˚C in 4% PFA/PBS, stained for 1 h in 1:2000 Hoechst/PBS, before 3x10 min washing steps in PBS. To determine the ratio of GFP expressing cells overview images were acquired in the Hoechst, GFP and Rhodamine channel using the 3x3 Snap functionality of the Axiovert 200 M fluorescence microscope (Zeiss) with a 63x (NA 1.3) oil objective at 103 nm pixel size. Uninduced cells acquired under identical conditions were used to determine the GFP signal threshold for positive cells. The cells that had not been used for imaging were treated in parallel with 0.1% Saponin/PBS. The isolated parasites were pelleted at 3300g for 5 min followed by a wash step and then boiled in 45 μl 1x Laemmli/PBS buffer at 95˚C. Hemozoin was removed via centrifugation at 21000g for 1 min. 15 μl of lysate was loaded onto 12% Mini-Protean TGX gels. All SDS-PAGEs were performed at 140 V for 45 min. Protein was then transferred onto a nitrocellulose membrane at 1.3 A, 25 V for 10 min using the Trans-Blot Turbo transfer system (Bio-Rad). The membrane was then blocked with filtered 5% milk/0.5% Tween-20/PBS blocking buffer for 15 min under agitation. Primary antibody staining occurred in the same buffer at 4˚C overnight under rotation. In addition to the rat anti-PfCen1 (this study) for staining of PfCen1, mouse anti-GFP (Roche #11814460001) and rabbit anti-PfAldolase (Abcam, #ab207494) were used for GFP and Aldolase staining respectively. Membranes were then washed 3x 10 min with blocking buffer under agitation. Secondary antibody incubation with anti-rabbit-680 (Rockland, #610-744-002), anti-rat-800 (Rockland, #612-145-002), or anti-mouse-800 (Rockland, #610-145-002) was performed in the dark at RT for 1 h under agitation. After 3 final washes with blocking buffer and once with PBS the membrane was analyzed using the LiCor Odyssey CLx system. Signal of bands corresponding to PfCen1-GFP and endogenous PfCen1 were quantified using the Image Studio Lite software v5.2 (LI-COR). The ratio was corrected for the share of the population not expressing PfCen1-GFP,quantified via microscopy as described above.

### Immunofluorescence assays

Immunofluorescence assays (IFA) of parasites for confocal and STED microscopy were performed as described previously in detail [83]. In brief, parasites were seeded to concanavalin A (Sigma-Aldrich) coated μSlide 8-Well glass bottom dishes (Ibidi), fixed the next day with 4% PFA (Electron Microscopy Sciences) in PBS (Gibco, Thermo Fisher Scientific) for 20 min at 37˚C, treated with 0.1% Triton X-100 (Sigma-Aldrich) and 0.1 mg/ml $NaBH_4$ (Sigma-Aldrich) PBS solutions for permeabilization and quenching of free aldehyde groups. Blocking as well as incubation of primary and secondary antibodies (S3 Table) was performed with 3% w/v Albumin Fraction V (Roth) in PBS. Hoechst 3342 (Thermo Fisher Scientific) was added during secondary antibody incubation at a dilution of 1:1000. Unbound antibodies were washed off with 0.5% Tween-20 (Roth) in PBS.

### Centrin foci frequency quantification

To quantify the frequency of ECCAs of PfCen1-GFP for different expression levels or mutants the respective seeded 3D7-DiCre_pFIO/pFIO+ strains were PFA fixed approximately 48 h post induction (or DMSO treatment) and an IFA for staining of DNA and tubulin was performed. In addition, GFP was stained using GFP-booster (ChromoTek gba488) at 1:200 to preserve GFP signal with minimal off-target staining if the sample was analyzed over multiple

imaging sessions. In case of quantifying the ECCA frequency of cells overexpressing PfCen1 without tag from pFIO+_GFP-T2A-Cen1 a staining with rat anti-PfCentrin1 was performed. For 3D7-DiCre_pFIO_Cen1-GFP, _pFIO+_Cen1-GFP, and _pFIO+_ΔN-PfCen1-GFP 54, 60, and 51 early schizonts (cells with intranuclear spindle and 1–5 nuclei) positive for GFP-foci were acquired in separate sessions. Foci outside the mitotic spindle poles were considered ECCAs.

For induced and DMSO treated 3D7-DiCre_pFIO+_GFP-T2A-Cen1 53 and 49 early schizonts, in case of the induced population positive for GFP-signal, were acquired. Due to the unspecific signals introduced by the anti-PfCentrin1 antibody an additional cutoff was applied and only foci outside of the mitotic spindle poles with a maximal signal that was at least as high as that of the weakest centrosomal foci were counted.

To determine the effect of 1,6-hexanediole on centrin foci DMSO treated (uninduced) 3D7-DiCre_pFIO+_GFP-T2A-Cen1 the medium of seeded cells was exchanged with pre-warmed cRPMI with 5% (v/v) 1,6-hexanediole for 5 min or left untreated (control), followed by a brief washing step in pre-warmed PBS and PFA fixation. IFA staining against DNA, tubulin, and PfCen1 was performed as described above. 36 1,6-hexanediole treated and 33 untreated early schizonts with at least one mitotic spindle were analyzed.

Individual cells were acquired using the Leica SP8 point laser scanning confocal microscope using an HC PL APO CS2 63x/1.4 N.A. oil immersion objective. 128x128 pixel multichannel images were acquired sequentially using HyD detectors in the standard mode with a pixel size of 72.6 nm and z-interval of 0.3 μm with a pinhole of 1 airy unit.

Analysis was performed using Fiji software [84]. For induced 3D7-DiCre_pFIO-EFh-dead-Cen1-GFP no quantification was performed as no cells with foci were ever found. GFP signal density was quantified using Fiji by applying a threshold (1, 255) to the GFP signal to isolate the cellular signal from the background. Area and raw integrated density of each individual Z-slice was determined via the multimeasure tool. The resulting intensity values were summed up and divided by the total area across all z-slices.

## Proteostat staining of ECCA positive cells

To determine if ECCAs are aggregates live seeded 3D7-DiCre_pFIO+_PfCen1-GFP cells were stained with 1:1000 Hoechst 3342 and 1:3000 Proteostat (Enzo Life Sciences) in cRPMI for 30 min at 37°C. Live cells were imaged at 37°C using the Leica SP8 microscope with the Lighting module, enabling automated adaptive deconvolution. 264x264 pixel images were acquired using a 63x/1.4 N.A. oil immersion objective, GaAsP detectors, a pinhole setting of 0.6 airy units, a pixel size of 35.1 nm and z-stack of 7.28 μm with slices at 130 nm intervals. PfCen1-GFP and Proteostat were excited with a 488 nm laser, with GFP emission being measured between 490 to 550 nm and Proteostat emission at 600 to 700 nm. GFP signal bleed-through into the Proteostat range was negligible.

## STED microscopy

STED microscopy of NF54-pARL_PfCen1-4 -GFP was performed on an Expert Line STED system (Abberior Instruments GmbH) equipped with SLM based easy 3D module and an Olympus IX83 microscopy body, using an Olympus UPlanSApo 100x oil immersion objective/1.4 NA with a pixel size of 20 nm. STED images on fixed cells were acquired in RescueSTED mode (to avoid structural damage from the STED laser heating up the parasite's hemozoin crystals) using the 590 and 640 nm excitation laser in line sequential mode with corresponding 615/20 and 685/70 emission filters. A 775 nm STED laser was employed at 10–15% intensity (maximum 3 W) with a pixel dwell time of 10 μs. Regular confocal images were

acquired in z-intervals of 300 nm, with 405 nm, 488 nm, 594 nm and 640 nm laser power being adjusted for each cell. Deconvolution of GFP staining signal was performed using Imspector Software (Abberior Instruments GmbH). For live cell STED seeded NF54-pARL_PfCen1-Halo were pre-treated with 1 μM SPY505-DNA (Spirochrome) and 1 μM MaP-SiR-Halo [37] for 3 h in cRPMI, which was replaced with cRPMI made from phenol red-free RPMI 1640 (PAN Biotech) prior to imaging. STED image series were acquired at 37˚C in 1–30 second intervals on a single Z-slice using the same laser setup as above. Deconvolution was performed with Huygens professional software using express deconvolution with the standard template.

## Quantification of live PfCen1-GFP dynamics during schizogony

For long term live cell imaging of seeded 3D7-DiCre_pFIO/pFIO+-PfCen1-GFP cells induced approximately 48 h prior were pre-incubated for 2 h at 37˚C in an airtight 8-well dish with 0.5 μM SPY650-Tubulin in cRPMI made from phenol red-free RPMI 1640 that has been pre-equilibrated in 3% $CO_2$, 5% $O_2$ overnight. Imaging was performed at 37˚C at the PerkinElmer UltraVIEW VoX microscope equipped with Yokogawa CSU-X1 spinning disk head and Nikon TiE microscope body with an Apo 60x/1.49 numerical aperture (NA) oil immersion objective and Hamamatsu C9100-23B electron-multiplying charge-coupled device (EM-CCD) camera with a pixel size of 59.8 nm. Images were acquired at multiple position using the automated stage and Perfect Focus System (PFS) in 5 min intervals. Multichannel images were acquired sequentially with 488 nm and 640 nm laser at 3% and 6% laser power as well as DIC. 8 μm stacks were acquired in 500 nm Z-intervals. For determination of the appearance of the first foci relative to the formation the first mitotic spindle the SPY650-tubulin signal was enhanced via deconvolution using the Fiji Deconvolutionlab2 with 10 iterations of the Richardson Lucy algorithm. PSF was generated using the PSF Generator plugin. Cells without visible focus at the start of imaging were visually analysed using Fiji. Mitotic spindle formation was identified by a strong increase in SPY650-tubulin foci signal and reduced mobility during the collapse of the hemispindle at the transition to a mitotic spindle.

For quantification of GFP signal within the cytoplasmic and foci fraction images were processed and analysed in a semi-automated workflow using ImageJ macros in the 64-bit 2.3.0 Fiji distribution of ImageJ. All used macros were deposited on https://github.com/SeverinaKlaus/ImageJ-Macros. Image segmentation was carried out in 3D using version 1.3.2 of the Windows distribution of the image classification software ilastik [85]. 3D timelapse GFP images were cropped to 80x80 pixel images containing individual iRBCs. Only iRBCs containing a single parasite were examined. To facilitate improved segmentation, image size was increased four times to 320x320 pixel using the adjust size function of ImageJ without interpolation. Segmentation of GFP foci was carried out on enlarged images using a pixel classification workflow in ilastik. With ilastik, a Random Forest classifier utilizing all features up to and including a sigma of 3.5 in 3D was trained on two reference GFP timelapses via manual annotation. Subsequently, all 3D GFP timelapse images were batch processed using the same classifier. Segmentation files were exported as multipage tiff and reconstructed into 4D image sets matching the original 80x80 files by reordering stacks and decreasing the image size via the size adjust function of ImageJ without interpolation. Reconstructed segmentation images were then converted into binary masks containing only 0 (background) and 1 (object, here GFP foci) pixel values using the ImageJ math functions. Subsequently, signal intensities for GFP were measured in the following compartments over time: the foci, the whole parasite excluding the foci and the whole parasite including the foci. In parallel, the total area occupied by each compartment at a given time was also quantified. Results were collected as comma separated

value (.csv) files and imported into Microsoft excel for analysis using an excel macro. To create masks for the whole parasite, images were segmented on background GFP values. Briefly, a median filter with a radius of 1 and a default threshold of 2200 (2200, 65535) were applied. To create binary parasite masks containing only 0 (background) and 1 (parasite) pixel values the thresholded image was divided by 255 using the ImageJ math functions. To create a mask of the whole parasite excluding the foci, the binary foci mask was inverted using the ImageJ math functions and the inverted foci mask multiplied with the parasite mask using the ImageJ image calculator function. The resulting binary mask contains only 0 (background and foci) and 1 (remaining parasite) pixel values. Areas were measured from binary segmentation masks by applying the multimeasure function on thresholded (1, 255) masks. To measure GFP signal intensities in different compartments, first background subtraction of 2000 was performed on the original 80x80 GFP images. The background subtracted image was then multiplied with one of the binary masks via the ImageJ calculator function, effectively removing any signal outside the compartment of interest. From the 4D image containing only the signal in the areas of interest, a summed z-projection was then created, from which signal intensity was measured as the raw integrated density using the ImageJ multimeasure function. Where indicated data of individual cells was normalized to the average value at the timepoints ranging from 10 min before and after mitotic spindle formation of that cell.

## Protein expression and purification

For the expression of PfCen1-6His variants as well as PfCen2-6His, and HsCen2-6His, chemically competent BL21-CodonPlus (De3) cells (Agilent Technologies) were transformed with the respective pET-28 vector via 42˚C heat shock and plated overnight. Due to low transformation efficiency a single colony was picked for inoculation of an overnight pre-culture with 50 mg/L kanamycin (Roth), which was diluted 1:50 the next day in pre-warmed LB with kanamycin. For the expression of PfCen3-6His, PfCen4-6His, and pZE13d-TEV-PfCen1-4 chemically competent W3110-Z1 *E. coli* [86] were transformed with the respective pZE13d vector via 42˚C heat shock and plated. The bacteria were scraped off and used to inoculate an expression culture at $OD_{600}$ of 0.06 in pre-warmed LB with 50 mg/L ampicillin (Roth). These expression cultures were grown at 37˚C, shaking at 130 rpm, until induction with 1 mM final IPTG (Thermo Scientific) once an $OD_{600}$ of 0.5 for W3110-Z1 or 0.7 for BL21 cells was reached, after which incubation continued for 3 and 5 h respectively. All further steps were performed at 4˚C. Bacteria were centrifuged and lysed via sonication (Branson Sonifier 250) in lysis buffer [3 mM β-Mercaptoethanol, 20 μg/ml DNaseI, 10 μg/ml Lysozyme, C0mplete protease inhibitor (Sigma-Aldrich), in PBS, pH 7.4]. Lysate was cleared via centrifugation at 17000 g for 20 min, supplemented with imidazole (Roth) to 10 mM, and the soluble fraction transferred to Ni-NTA beads (QIAGEN) in a gravity flow column. Beads were washed with a 10x volume of wash buffer [50 mM $NaH_2PO_4H_2O$, 300 mM NaCl, pH 8.0] at 20 mM imidazole and protein eluted at 250 mM imidazole. In case of 6His-TEV-PfCen1-4, 6His-TEV protease (EMBL Protein Expression and Purification Core Facility) was added to cleave of the 6His tag (leaving an N-terminal glycine) and buffer exchanged to PBS with 3 mM β-Mercaptoethanol via consecutive 2 h and overnight dialysis in a 300x volume using a 6–8 kDA cut-off tubing (Spectra/Por, 132650) at 4˚C. Another round of reverse Ni-NTA purification was then performed to remove the 6His-TEV from the solution, collecting the flowthrough. With these and all other proteins buffer was exchanged to 50 mM BisTris pH 7.1 using the same dialysis protocol. Aliquots were snap-frozen in liquid nitrogen for storage at -80˚C and slowly thawed on ice prior to experiments. Purity was determined via Coomassie gel. Concentrations were determined via Pierce 660 nm protein assay (Thermo Fisher) using pure bovine serum albumin (Thermo Fisher) as a

standard. Purity was confirmed by loading 1.125 μg of protein on a 12-well 12% Mini-Protean TGX gel (Bio-Rad), performing SDS-PAGE electrophoresis and fixing the protein with 40% methanol, 10% acetic acid solution, and staining with 0.25% Coomassie-blue (Merck) in 50% methanol, 10% acetic acid. Gels were documented with the Li-COR Odyssey CLx (Li-COR Biosciences) at 700 nm. Correct protein sizes were confirmed via LC-MS by the CFMP at the ZMBH Heidelberg, using a maXis UHR-TOF (Bruker Daltonics) coupled with nanoUPLC Acquity (Waters) tandem mass spectrometer for intact protein detection.

### Antibody generation

For generation of polyclonal rat anti-PfCentrin1, PfCentrin1-6His was additionally purified via size exclusion chromatography as described previously [12]. Immunization was performed by Davids-Biotechnology using a 250 μg, 63 day, 5 injection immunization regimen. Anti-PfCen1 rat serum was used for all assays, without further affinity purification.

### LLPS quantification and imaging

To quantify LLPS over time 350 μl of protein sample solutions in 50 mM BisTris buffer at pH 7.1 were prepared in a 1.5 ml polystyrene cuvette (Sarstedt) on ice and transferred into a NP80 nanophotometer (Implen), which was blanked with buffer and preheated to 37˚C. In case of PfCen1-GFP-6His the protein solution itself was used for blanking due to absorption of GFP. The $OD_{200-900}$ 10 mm pathlength was then measured in 10 s intervals for 25 min, or 1 min intervals for up to 3 h in case of long-term measurements. At 450 s 3.5 μl 200 mM $CaCl_2$ or $MgCl_2$ solution was added to 2 mM (first 330 s not shown), and at 1170 s 7 μl 500 mM EDTA solution to 10 mM. To ensure even distribution of the solutions during addition into the cuvette they were pre-loaded into a 200 μl pipette tip, which was used to pipette sample solution up and down three times in between two measurements. To measure the effect of 1,6-hexanediole 39 μl of a 50% (v/v) solution in $ddH_2O$ was added at 1170 s in the same manner. To easily detect droplet fusion events, we used increased centrin concentrations for the movies (PfCen1-6His, 200 μM. PfCen2-6His, 120 μM. PfCen3-6His, 193 μM. PfCen4-6His, 120 μM. HsCen2-6His, 200 μM). Otherwise, 30 μM was used or as indicated. 20 μl protein sample were imaged in non-binding PS μCLEAR 384 well plates (Greiner). For fluorescent imaging the 30 μM protein solutions were supplemented with 300 nM (100:1 molar ratio) NTA-Atto 550 (Sigma Aldrich, Merck). Movies were acquired at 37˚C with a PerkinElmer UltraVIEW VoX microscope equipped with Yokogawa CSU-X1 spinning disk head and Nikon TiE microscope body with an Apo 60x/1.49 NA oil immersion objective and Hamamatsu sCMOS OrCA Flash 4.0 camera with a pixel size of 89.7 nm. Image series were taken in 1 s intervals using differential interference contrast (DIC) and 561 nm laser. For addition of 0.2 μl 200 mM $CaCl_2$ solution or 0.4 μl 500 mM EDTA solution to the well the plate was briefly removed from the microscope for access. To assess co-condensation 150 μl of a 30 μM PfCen3-6His was imaged inside an untreated μSlide 8-Well glass bottom dishes (Ibidi), induced with 1.5 μl 200 mM $CaCl_2$ solution, and supplemented during imaging with 20 μl PfCen1-GFP-6His directly into the well to a final concentration of 30 μM. GFP was excited using a 488 nm laser.

### Analysis of N-terminal IDRs in centrins

Likelihood of IDRs was estimated using the IUPred3 tool (www.iupred3.elte.hu) [87]. Centrins with a ≥50% likelihood for N-terminal IDRs were considered positive. The phylogenetic tree was generated using ClustalOmega (www.ebi.ac.uk/Tools/msa/clustalo/), using default settings.

## Circular dichroism spectroscopy of recombinant centrins

Recombinant wild type Cen1-6His or mutant EFh-dead PfCen1-6His was diluted with size exclusion chromatography (SEC) buffer, 50 mM Tris-HCl pH 7.1 at 4˚C, to 0.4 ml. The samples were injected onto a Superdex S200 10/300 column (equilibrated in SEC buffer) running at a flow rate of 0.75 ml/min in the SEC buffer. Samples from the different fractions were loaded on 4–20% Mini-PROTEAN TGX Precast Gels (Bio-Rad, 4561095) and run under denatured and reduced conditions using Tris/glycine/SDS running buffer (Bio-Rad, 1610772). The gel was run for 15–17 min at 300 V and stained with Coomassie Blue. Fractions with highly purified protein were diluted into the SEC buffer to a final concentration of 0.5 mg/ml and centrifuged for 10 min at maximal speed prior to the circular dichroism (CD) measurements. CD spectra were recorded using a Jasco J-815 CD spectrophotometer using a 0.2 mm path-length CD Quartz cuvette, at 20˚C, between 260 nm to 190 nm, with 50 nm/min scan speed, digital integration time of 1 sec, a bandwidth of 1 nm, a data-pitch of 0.1 nm and 10 accumulations were averaged. A buffer spectrum was first recorded as baseline and subtracted from the sample spectrum. The spectra were analysed with the Contin-LL [88] method with the protein reference data 7 [89] using the Dichroweb server [90].

## Statistics

Statistical significance of the difference in the timepoint of PfCen1-GFP foci appearance between pFIO and pFIO+ (Fig 4F) was assessed via a two-tailed t-test with n = 46 and n = 45 respectively. Significance of difference in centrosomal PfCen1 foci size in induced and control cells (S9 Fig) was assessed using a t-test with Welch's correction with n = 31 and 23 respectively.

## Supporting information

**S1 Fig. Schematic map of pARL vector used for ectopic expression of tagged centrins.** The pARL vector contains a hDHFR cassette, conferring resistance to antifolates, and drives expression of a gene or fusion gene of choice from a weakened promoter of the *P. falciparum* chloroquine resistance transporter (Pfcrt).
(TIF)

**S2 Fig. Recombinantly produced centrins have full length size.** (A) Coomassie gels of wild type centrins recombinantly produced in bacteria and purified via their C-terminal 6His tag show slightly lower migration than expected for PfCen2-6His and PfCen-4-6His (B) Mass spectrometry analysis of native proteins, however, confirms full length expression for all wild type centrins except for cleavage of methionine 1 and occasional detection of degradation bands. (C) Coomassie gels of mutant, tagged and tag-free centrin versions confirms proper expression and show expected migration. Non-plasmodium centrins are *T. brucei* centrin A (Accession number: XP_846945) 22 kDa, *C. reinhardtii* centrin (Accession number: P05434) 20 kDa, and S. cerevisiae centrin (Cdc31) 19 kDa.
(TIF)

**S3 Fig. Centrin phase-separation also occurs at lower concentrations and is calcium specific.** (A) Transmission light images of recombinant PfCen1-6His, PfCen3-6His, and HsCen2-6His centrin protein solutions at 30 µM concentration before, after calcium, and EDTA addition. Proteins were fluorescently labeled via their 6His tag using 300 nM NTA-Atto 550. Time stamps indicate time elapsed between calcium or EDTA addition and image acquisition. Observed droplets are more sparse than in highly concentrated solution but are still dynamic. (B) Turbidity assay using addition of magnesium followed by calcium addition at the same

concentration indicates $Ca^{2+}$ specifically. and not bivalent cations per se, induces centrin LLPS. (C) Testing different PfCen1-6His concentrations reveals saturation concentration to be below 15 μM. (D) Turbidity assay with recombinant PfCen1-GFP at 20 μM. (E) Turbidity assay with recombinant PfCen1-4 after proteolytic cleavage of any protein tag using various concentrations as indicated. Scale bar: 10 μm. Conditions: 50 mM BisTris (pH 7.1), addition of $CaCl_2$ or $MgCl_2$ to 2 mM and EDTA to 10 mM, 37˚C.
(TIF)

**S4 Fig. Non-reversible condensate fraction of PfCen3 increases over time.** (A) Turbidity of PfCen3-6His solution with varying time points of EDTA addition after Calcium addition. (B) Turbidity of PfCen1-6His protein solution with highly delayed EDTA addition shows no irreversible fraction. Overall reduction of turbidity could be explained by protein droplets settling down. Conditions: 50 mM BisTris (pH 7.1), addition of $CaCl_2$ to 2 mM and EDTA to 10 mM, 37˚C.
(TIF)

**S5 Fig. Phase-separating centrins contain intrinsic disordered regions in their N-terminus.** (A) Using the IUpred 3.0 prediction software we identified a higher intrinsic disorder region probability (red line) in centrins undergoing LLPS in vitro. The highest probabilities are found in the N-terminus before the EFh domains. (B) Using Clustal Omega provided by the EMBL European Bioinformatics Institute we created a phylogenetic tree of centrins from multiple highly divergent eukaryotes, for several of which centrin self-assembly has been shown in vitro in this and previous studies (*). Using IUpred 3.0 we determined centrins with an IDR probability above the 0.5 threshold (red) or without increased IDR probability (blue). Centrins analyzed in this study are in bold.
(TIF)

**S6 Fig. PfCen1 and 3, but not PfCen2 and 4 contribute to co-condensation.** Turbidity measurements of recombinant centrins at individually subcritical concentrations either independently or as a mixture during addition of calcium followed by EDTA as indicated. All conditions: 50 mM BisTris (pH 7.1) at 37˚C.
(TIF)

**S7 Fig. pFIO constructs enable robust inducible overexpression of PfCen1-GFP in malaria parasites.** (A) Flow cytometry analysis of multinucleated malaria parasites (stained with SYTO 61 as a DNA marker) expressing PfCen1-GFP using the classical pARL construct only detects a small fraction of GFP-positive cells with a low median fluorescence intensity. (B) Fraction of GFP-positive parasites strongly increases upon induction of pFIO-PfCen1-GFP transfected cells by rapamycin addition. Median GFP fluorescence intensity is much higher than in pARL whereas the hsp86 promoter generates even higher values than the hsp70 promoter fragment.
(TIF)

**S8 Fig. Overexpression levels of pFIO+_PfCen1-GFP.** Western blot analysis of purified 3D7-DiCre_pFIO+_PfCen1-GFP late-stage parasites treated with rapamycin or DMSO (control) in the previous cycle. The signal of endogenous centrin (19.6 kDa, 93 AU) and PfCen1-GFP (46.5 kDa, 717 AU) stained with rat anti-PfCen1 was quantified. Corrected for PfCen1-GFP being only present in a subset of the population (43%, n = 662), as quantified via microscopy, an average (induced) cell has an approximate PfCen1:PfCen1-GFP ratio of 1:18. Staining with anti-PfAldolase was employed as a loading control. Anti-GFP staining indicated no cleavage of PfCen1-GFP, with no expression being detected in the DMSO treated populations with either staining. An additional band at ~21 kDa in the anti-PfCen1 staining is likely

the result of cross-reactivity with PfCen3 (20.9 kDa).
(TIF)

**S9 Fig. ECCA appearance correlates with Cen1-GFP expression levels.** (A) Graph shows total arbitrary GFP fluorescence intensity of cells transfected with pFIO/pFIO+_PfCen1-GFP one asexual cycle after induction, plotted against the fraction of centrin foci, which are ECCAs per centriolar plaque associated centrin foci. Despite the low r-squared value statistical analysis indicate that the slope is significantly not zero (p<0.0001) and therefore positively correlated. (B) as in A but measured for pFIO+_PfCen1-GFP expressing cells. Slope of linear regression is also significantly not zero (p<0.033) as determined by an F-test. Due to the different expression levels different excitation laser settings had to be used for both parasite lines.
(TIF)

**S10 Fig. Dimensions of the anti-PfCen1 signal at the centriolar plaque in overexpressing parasites.** Quantification of immunofluorescence signal at the centriolar plaque measured as in Fig 1D–1E using anti-PfCen1 labeling in rapamycin-induced parasites overexpressing PfCen1-GFP vs control. Using t-test with Welch's correction width (321 vs 324 nm) shows no difference between the conditions while the height of the signal (247 vs 219 nm) was statistically higher in the induced parasites (p = 0.0081).
(TIF)

**S11 Fig. Centrin signal concentration in foci is constant prior to spindle formation.** Mean PfCen1-GFP fluorescence intensity by segmented cellular region in induced parasites carrying pFIO+_PfCen1-GFP relative to mitotic spindle formation in time lapse movies. N > 32.
(TIF)

**S12 Fig. T2A skipping is very efficient in pFIO+_GFP-T2A-PfCen1.** Western blot analysis of protein extract from rapamycin-induced and control late-stage parasites carrying pFIO+_GFP-T2A-PfCen1 plasmid. Anti-GFP antibody was used for detection and anti-PfAldolase as a loading control. Rapamycin-induced parasite extract shows a clear band at the expected size (arrow) for "skipped" GFP while the band for the "unskipped" product is barely detectable.
(TIF)

**S13 Fig. Crossreactivity of rat anti-PfCen1 antibody.** Western blot analysis using newly generated rat anti-PfCen1 antibody. Antibody was tested against total parasite protein extract of late stage wild type NF54 (left lane) and recombinant PfCen1-4-6His proteins (right lanes). Aside some unspecific background the antibody detects a band of the expected size (20 kDa) in parasite extract. Anti-PfCen1 strongly crossreacts with recombinant PfCen3-6His. Two different contrast adjustments of the same blot are shown.
(TIF)

**S14 Fig. PfCen1 N-terminus is essential but not sufficient for centrin phase separation.** (A) Turbidity of chimeric centrin fused from disordered N-terminus of PfCen1 and EFh domain containing part of PfCen4 during calcium and EDTA addition to protein solution. (B) Turbidity of chimeric centrin fused from N-terminus of PfCen4 and EFh domain containing part of PfCen1 during calcium and EDTA addition to protein solution. Conditions: 50 mM BisTris (pH 7.1), addition of $CaCl_2$ to 2 mM and EDTA to 10 mM, 37°C.
(TIF)

**S15 Fig. EFh-dead point mutations have no substantial effect on the secondary structures of PfCentrin1 protein.** CD spectra of recombinant PfCen1-6His (black) and EFh-dead

PfCen1-6His mutant (magenta) protein show no difference. Calculated secondary structure fractions from the CONTIN-LL analysis method of wild type PfCen1 vs mutant PfCen1 yielded 24.7% vs 24.6% helices, 23.0% vs 22.9% strands, 15.8% vs 15.9% turns, and 36.5% vs 36.6% unordered regions, respectively.
(TIF)

**S1 Table. Primers used in this study.** All primers used for amplification of the various construct fragments used for molecular cloning as described above are listed here. Homology regions for Gibson Assembly are in lowercase, PCR binding sequences are capitalized. * Kindly provided by Dr. Friedrich Frischknecht.
(PDF)

**S2 Table. Oligos used in this study.** Lowercase letters indicate Gibson assembly or ligation overhangs. All listed ds DNA oligos used for molecular cloning were ordered as GeneArt Strings (ThermoFisher).
(PDF)

**S3 Table. Antibodies used in this study.** List of antibodies with information of species, final dilution and source with order number. Starred (*) are dilutions for confocal IFAs. For STED microscopy concentration was increased to 1:200.
(PDF)

**S1 Movie. Some PfCen1-Halo foci display round fluid-like dynamics.** STED time lapse movie at 1 s time interval of centriolar plaque region of parasites expressing PfCen1-Halo labeled with MaP-SiR-Halo dye (green). DNA stained with SPY505-DNA (blue). Scale bar, 100 nm. Linear bleaching correction was applied using Fiji.
(AVI)

**S2 Movie. Some PfCen1-Halo foci display stretched fluid-like dynamics.** STED time lapse movie at 1 s time interval of centriolar plaque region of parasites expressing PfCen1-Halo labeled with MaP-SiR-Halo dye (green). DNA stained with SPY505-DNA (blue). Scale bar, 100 nm. Linear bleaching correction was applied using Fiji.
(AVI)

**S3 Movie. PfCen1 displays LLPS upon calcium addition in vitro.** Transmission time lapse imaging of recombinant PfCen1-6His protein solution at 200 μM concentration displays wetting and fusion events rapidly after calcium addition. 50 mM BisTris (pH 7.1), addition of $CaCl_2$ to 2 mM and EDTA to 10 mM, 37˚C. Image dimensions are 50 x 50 μm.
(AVI)

**S4 Movie. PfCen3 displays LLPS upon calcium addition in vitro.** Transmission time lapse imaging of recombinant PfCen3-6His protein solution at 193 μM concentration displays wetting and fusion events rapidly after calcium addition. 50 mM BisTris (pH 7.1), addition of $CaCl_2$ to 2 mM and EDTA to 10 mM, 37˚C. Image dimensions are 50 x 50 μm.
(AVI)

**S5 Movie. HsCen2 displays LLPS upon calcium addition in vitro.** Transmission time lapse imaging of recombinant HsCen2-6His protein solution at 200 μM concentration displays wetting and fusion events rapidly after calcium addition. 50 mM BisTris (pH 7.1), addition of $CaCl_2$ to 2 mM and EDTA to 10 mM, 37˚C. Image dimensions are 50 x 50 μm.
(AVI)

**S6 Movie. PfCen3 droplets are not fully resolved by EDTA and display more solid-like properties.** Transmission time lapse imaging of recombinant PfCen3 protein solution at 193 μM concentration after calcium and EDTA addition does not dissolve droplets. Remaining spheres don't display fusion or wetting anymore. 50 mM BisTris (pH 7.1), addition of CaCl$_2$ to 2 mM and EDTA to 10 mM, 37˚C. Image dimensions are 50 x 50 μm.
(AVI)

**S7 Movie. PfCen3 displays LLPS upon calcium addition in vitro at temperatures below 37˚C.** Transmission time lapse imaging of recombinant PfCen3-6His protein solution at 193 μM concentration displays wetting and fusion events rapidly after calcium addition for longer at lower temperatures. 50 mM BisTris (pH 7.1), addition of CaCl$_2$ to 2 mM and EDTA to 10 mM, 32.5˚C. Image dimensions are 50 x 50 μm.
(AVI)

**S8 Movie. Calcium-induced TbCenA droplets are fusogenic.** Transmission time lapse imaging of recombinant TbCenA-6His protein droplets in solution at 100 μM concentration displays fusion events after calcium addition. 50 mM BisTris (pH 7.1), addition of CaCl$_2$ to 2 mM, 37˚C. Time lapse is 1 second per frame. Image dimensions are 10 x 10 μm.
(AVI)

**S9 Movie. Calcium-induced ScCdc31 droplets display surface wetting.** Transmission time lapse imaging of recombinant ScCdc31-6His protein droplets in solution at 200 μM concentration displays surface after calcium addition. 50 mM BisTris (pH 7.1), addition of CaCl$_2$ to 2 mM, 37˚C. Time lapse is 1 second per frame. Image dimensions are 10 x 10 μm.
(AVI)

**S10 Movie. Calcium-induced CrCen droplets display surface wetting.** Transmission time lapse imaging of recombinant CrCen-6His protein droplets in solution at 100 μM concentration displays surface after calcium addition. 50 mM BisTris (pH 7.1), addition of CaCl$_2$ to 2 mM, 37˚C. Time lapse is 1 second per frame. Image dimensions are 10 x 10 μm.
(AVI)

**S11 Movie. PfCen1-GFP overexpression causes ECCA formation.** Live cell spinning disk time lapse movie at 5 min time intervals of parasite overexpressing PfCen1-GFP (green) from pFIO+ labeled with SPY650-Tubulin (magenta) during entry into schizogony at 37˚C. Image dimensions are 15 x 15 μm. Left frame only shows PfCen1-GFP signal. Maximum intensity projection.
(AVI)

**S12 Movie. ECCAs frequently form at centriolar plaques before detaching.** Short live cell spinning disk time lapse movie at 5 min time interval of schizont parasite overexpressing PfCen1-GFP (green) from pFIO+ labeled with SPY650-Tubulin (magenta) showing detachment of the brightest foci from the spindle. Image dimensions are 15 x 15 μm. Left frame only shows microtubule signal. Maximum intensity projection.
(AVI)

## Acknowledgments

We thank: The Infectious Diseases Imaging Platform for imaging support and use of their microscopes (idip-heidelberg.org). PlasmoDB.org for their *Plasmodium* Informatics Resources. Core Facility for Mass Spectrometry & Proteomics (ZMBH). Nicolas Lardon and Kai Johnsson (Max Planck Institute for Medical Research) for providing the MaP-SiR-Halo

dye. Anthony Holder and Moritz Treeck for the 3D7 DiCre strain. Marina Iocca for help with molecular cloning and protein expression. Arne Börgel and Karine Lapouge from the EMBL-Heidelberg Protein Expression and Purification Core Facility for carrying out size exclusion chromatography and CD spectroscopy analysis. Friedrich Frischknecht, Sebastian Baumgarten, and Daniel Gerlich for critical discussion of the manuscript.

## Author Contributions

**Conceptualization:** Yannik Voß, Julien Guizetti.

**Data curation:** Yannik Voß.

**Formal analysis:** Yannik Voß.

**Funding acquisition:** Markus Ganter, Julien Guizetti.

**Investigation:** Yannik Voß, Severina Klaus, Nicolas P. Lichti.

**Methodology:** Yannik Voß, Severina Klaus.

**Project administration:** Julien Guizetti.

**Resources:** Yannik Voß, Nicolas P. Lichti.

**Supervision:** Markus Ganter, Julien Guizetti.

**Validation:** Yannik Voß.

**Visualization:** Yannik Voß, Julien Guizetti.

**Writing – original draft:** Julien Guizetti.

**Writing – review & editing:** Yannik Voß, Markus Ganter.

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
