## [Decision Letter · Decision Letter 0]

4 Nov 2023

Dear Dr. Guizetti,

Thank you very much for submitting your manuscript "Malaria parasite centrins can assemble by Ca2+-inducible condensation" for consideration at PLOS Pathogens. As with all papers reviewed by the journal, your manuscript was reviewed by members of the editorial board. This manuscript was previously assessed by Reviews Commons, and the reviewers appreciated the attention to an important topic but raised some  concerns. We have opeted to sent  the revision back to the same reviewers.  Based on the reviews, we are likely to accept this manuscript for publication, providing that you address the outstanding major and minor points and modify the manuscript according to the review recommendations.

Sincerely,

Dominique Soldati-Favre

Section Editor

PLOS Pathogens

Dominique Soldati-Favre

Section Editor

PLOS Pathogens

Kasturi Haldar

Editor-in-Chief

PLOS Pathogens

orcid.org/0000-0001-5065-158X

Michael Malim

Editor-in-Chief

PLOS Pathogens

orcid.org/0000-0002-7699-2064

Reviewer Comments (if any, and for reference):

Reviewer's Responses to Questions

**Part I - Summary**

Reviewer #1: The revision of this submission has greatly improved it; the issues raised in my initial review have been satisfactorily addressed and I consider that the data in the paper now support the major conclusions drawn in the study. The expanded discussion positions this work in the context of the literature and will allow readers to consider the ramifications of the authors’ findings in several different directions. I think this paper makes a valuable and interesting contribution to centrosome biology.

Reviewer #2: Plasmodium falciparum survival is very much dependent on its ability to efficiently replicate and one important component of the nuclear replication machinery is centrin. Understanding this can help identify novel drug targets against a parasite that is well known to have developed resistance to almost all anti-malarials. The study has extensively characterised centrins in P. falciparum. The authors have described centrin in normally replicating parasites as well as in the case of protein overexpression in vivo, both instances providing important insights on centrin. Although episomally expressed tagged proteins were used due to the difficulty of tagging the endogenous protein, necessary controls were done, further supporting the data being presented here. The study also covered in vitro characterisation of centrin, demonstrating LLFS in the presence of calcium (but not magnesium), and reversed in the presence of EDTA. They have also conducted similar experiments involving centrins from other evolutionarily distant eukaryotes with predicted intrinsic disordered regions. This therefore will be of great interest to a wider audience, and not just in the malaria community or parasitology. Perturbations on the centrin protein (i.e., removal of the N-terminal intrinsic disordered regions, EFh mutation) demonstrated the importance of these protein regions in liquid-liquid phase separation. The authors have provided circular dichroism spectroscopy to ensure that the mutations do not affect the protein’s secondary structure.

Reviewer #3: The authors conducted several experiments and analyses to address the comments raised by the three reviewers of the initial submission. The manuscript is clearly improved as a result. In my opinion, however, a few outstanding issues remain to be clarified before publication could be endorsed, as detailed hereafter.

**Part II – Major Issues: Key Experiments Required for Acceptance**

Reviewer #1: (No Response)

Reviewer #2: No major issues

Reviewer #3: Major points

The major remaining outstanding issue pertains to the relevance of these observations to the behavior of PfCen1 and PfCen3 in vivo, an issue highlighted also by Reviewer 1 and to some extent by Reviewer 3 in their assessment of the initial submission. Obviously, the best way to address this would be to endogenously tag PfCen1 or PfCen3, which the authors argue they cannot do (although they did not attempt N-terminal tagging). In the absence of endogenous tagging, could the authors provide cells with plasmids where PfCen1-GFP or PfCen3-GFP are driven by their respective endogenous promoters? Without such experiments, one is left wondering whether ECCAs might simply result from proteins being overexpressed at non-physiological levels. The authors argue that overexpression experiments are by design (“ the overexpression is, however, used intentionally since protein concentration correlates with the phase separation”), but given the very weak correlation in question (see Fig. S8), perhaps overexpression in all cases is well above the levels of endogenous components, which would then bring little understanding to how these proteins function in regular cell physiology. Another way to mitigate this critical concern would be to show that the level of overexpressed PfCen1-GFP or PfCen3-GFP approximates the levels of the corresponding endogenous proteins. Despite the arguments presented by the authors, it is not clear why they could not estimate the extent of overexpression using PfCen1 antibodies, despite the cross-reactivity towards PfCen3. For instance, they could conduct semi-quantitative immunoblot analysis, taking into account that only ~50% of cells are GFP-positive (see Fig. S7), or else perform immunofluorescence analysis. Despite the cross-reactivity with PfCen3, this should allow them to place a lower bound on the extent of overexpression. If variability of expression levels between cells is an issue, these analyses could be conducted following FACS sorting. Moreover, the authors could also envisage conducting semi-quantitative proteomic analysis, should they want to ascertain this question beyond doubt.

One argument the authors use to argue that overexpression does not impact cell physiology is the absence of significant growth phenotype (see response to Reviewer 1: “Further, we found no significant growth phenotype in overexpressing parasites, which indicates that the centriolar plaque is functional.”). However, in response to Reviewer 3, they write that the “parasite lines seem to silence the Cen1-4-GFP expression plasmids readily, which suggests that there might be a growth disadvantage”, mentioning also that they could to assess this with certainty. Taken together, these comments leave me confused regarding whether there is a growth disadvantage or not. Also, if variability in GFP expression levels is an issue to address this with certainty, the authors could sort cells using FACS and analyze the growth of cells as a function of GFP levels. Overall, this point is important to clarify because a growth defect would be indicative of overexpression interfering with cell physiology.

In response to my comment B), the authors argue that Movies S1 and S2 are not meant to demonstrate the presence of biomolecular condensates, and that the focal planes in Movies S3 and S4 have been selected to highlight the hallmarks of LLPS. What I see in Movie S3 is that the few droplet fusion events (e.g. at 9 seconds into the movie) are, in fact, out of focus. From their reply (“… throughout the entire imaged liquid volume…”), I assume that the authors acquired Z stacks, so that they should choose better focal planes to illustrate such fusion events. As for Movie S4, it appears rather to show that droplets do NOT fuse, as many that appear in physical proximity do not coalesce (again, the fact that the focal plane is usually not the best makes it difficult to say for sure), despite PfCen3 being present at 200 microM. This would argue against LLPS occurring in this case. Could it be that PfCen1 and PfCen3 differ in this respect? Also, a key expectation of LLPS is that droplet size should increase over time. Whereas this might be happening with PfCen1 (although it is difficult to say for sure in the absence of quantification), this does not appear to be the case for PfCen3. The important points must be clarified by further analyses.

**Part III – Minor Issues: Editorial and Data Presentation Modifications**

Reviewer #1: I have a couple of minor points to tidy up before final publication.

1. The reference list should be reviewed for typos, omissions, repetitions etc. There are a lot of mistakes- one presumes this is due to the referencing software, but it is important to get this right in the final version.

2. Some minor typos:

Line 156 ‘evolutionarily’

Line 188 (and 435, 446, 447, 456): ‘to condensate’ is not a verb. It should be replaced by ‘to form a condensate’, ‘to demonstrate condensation’, ‘to undergo condensation’ etc.

Line 217 ‘reevaluating our’

Line 244 ‘pFIO where GFP’

Line 257 ‘not dependent on’

Line 469 ‘attributed’

Line 880 ‘n =’ (I presume).

Reviewer #2: 1. Line 217: “When reevaluating <of> our earlier data from the weaker expressing pArl-Cen1/3-GFP parasite lines…”

Remove “of”

2. Line 243: “To exclude that ECCAs only form due to GFP-tagging we designed a version of pFIO were GFP and PfCen1 were separated by a T2A skip peptide (Fig 5H).”

Should have a “+” after pFIO

3. Figure 5H

Promoter in the figure should only show hsp86 (instead of hsp 70 or 86) to represent pFIO+

4. Line 293: “Taken together these observations argue for  condensation being part of the mechanism of centrin assembly at the centriolar plaque.”

Remove “a”

5. Line 581, 600, 638, 681, 741 …

Update referencing format

6. Line 674: “ IFA staining against…”

Remove “A”

7. Line 690: <li> Live cells

Change “Life” to “Live”

8. Line 693: “264x264 pixel images were acquired L HC PL APO CS2 63x/1.4 N.A. oil immersion objective was employed with GaAsP detectors, a pinhole of 0.6 airy units, a pixel size of 35.1 nm and z-stack of 7.28 μm at 130 nm intervals.”

Kindly rephrase

9. Standardise use of “6His” vs “6his” vs “6xHis”

10. Figure S3: missing caption/legend for S3E</life></of>

Reviewer #3: Other points

In an important experiment, the authors tested whether overexpression of untagged PfCen1 also yields ECCAs, finding that this is the case in ~50% of cells in the control (DMSO) condition, and ~80% of cells upon rapamycin addition. Why do ~50% of control cells already have ECCAs? Is this is due to leaky expression? If so, why is this not the case with the constructs inducing PfCent1-GFP expression?

In response to comment 7 of my initial review, the authors write that their data do not conflict with the previously reported notion that Centrin is part of a helical inner scaffold, and mention that they have discussed this matter with the corresponding authors of the study in question, who are said to be in agreement with them. This is surprising in light of what is stated in that manuscript (e.g. the abstract states “POC5, POC1B, FAM161A, and Centrin-2 localize to the scaffold structure along the inner wall of the centriole MTTs.”).

The authors write (lines 83-84) that “our experience confirms that tagging of the C-terminus of PfCen1 and 3 is not achievable even using small tags”, they should spell out in some more detail what they tried (e.g. in the Materials and methods section). Otherwise this information is of limited use to the community.

The authors write (lines 121-123) that “the critical saturation concentration for PfCen1-GFP is well within the range of what can be found for in vitro phase separation of other proteins”, referring to the Li et al, 2020 manuscript. However, an exploration of the LLPSDB data base reported in that manuscript does not seem to provide such values. This should be clarified, perhaps by providing a few exemplary values for readers to appreciate this point.

The authors write (lines 286-287) that the fact that the dissolving effect of hexanediol of PfCen1 in vitro was moderate could be attributed to the polar components of the protein . Is this this also the case for PfCen3? Regardless, how do the authors imagine that the postulated impact of such polar components is shielded in vivo?

The authors write (lines 292-293) that “We further detected a disorganization of the mitotic spindles, which seemed longer and less dense.” This is not apparent from the figure panels. The authors should highlight these feature and, to the extent possible, quantify them.

Line 358-359: the authors mention that Centrins are conserved in all eukaryotes analyzed. To my knowledge, there are no Centrins in nematodes such as C. elegans, nor in insects such as Drosophila. The writing should be rectified to reflect this fact.

In the Material and methods (lines 817-821), the authors report how PfCen1 antibodies were raised, but not how they were affinity-purified. This information must be provided, also for better understanding why these antibodies cross-reacts with PfCen3 but not with the other two Centrin proteins.

PLOS authors have the option to publish the peer review history of their article (what does this mean?). If published, this will include your full peer review and any attached files.

Reviewer #1: No

Reviewer #2: No

Reviewer #3: No

Figure Files:

Data Requirements:

Reproducibility:

References:

---

## [Editor Report · Decision Letter 1]

13 Dec 2023

Dear Dr. Guizetti,

We are pleased to inform you that your manuscript 'Malaria parasite centrins can assemble by Ca2+-inducible condensation' has been provisionally accepted for publication in PLOS Pathogens.

However there are two outstanding points that  the reviewer recommends to address

1. It should be more explicitly indicated in the text that ECCAs  are form upon vast overexpression (now finally estimated to be ~18x the endogenous levels), and that this impacts parasite fitness. 

2.  The authors  responded that centrin is called “caltractin” in worms and flies, but there are no such proteins and, to my knowledge, centrins are indeed lacking in nematodes and insects. 

Before your manuscript can be formally accepted you will need to complete some formatting changes, which you will receive in a follow up email. A member of our team will be in touch with a set of requests

Best regards,

Dominique Soldati-Favre

Section Editor

PLOS Pathogens

Dominique Soldati-Favre

Section Editor

PLOS Pathogens

Kasturi Haldar

Editor-in-Chief

PLOS Pathogens

orcid.org/0000-0001-5065-158X

Michael Malim

Editor-in-Chief

PLOS Pathogens

orcid.org/0000-0002-7699-2064
---

## [Editor Report · Acceptance letter]

20 Dec 2023

Dear Dr. Guizetti,

We are delighted to inform you that your manuscript, "Malaria parasite centrins can assemble by Ca2+-inducible condensation," has been formally accepted for publication in PLOS Pathogens.

Best regards,

Michael Malim

Editor-in-Chief

PLOS Pathogens

orcid.org/0000-0002-7699-2064